

# Revisiting the Relationship between Atlantic Dust and Tropical Cyclone Activity using Aerosol Optical Depth Reanalyses: 2003-2018

Peng Xian[1a], Philip J. Klotzbach[2a], Jason P. Dunion[3], Matthew A. Janiga[1], Jeffrey S. Reid[1] , Peter R. Colarco[4] and Zak Kipling[5]

[1]Naval Research Laboratory, Monterey, CA, USA.
[2]Department of Atmospheric Science, Colorado State University, Fort Collins, CO, USA.
[3]University of Miami/RSMAS/CIMAS – NOAA/AOML/Hurricane Research Division, Miami, FL, USA.
[4]NASA Goddard Space Flight Center, Greenbelt, MD, USA.
[5]European Centre for Medium-Range Weather Forecasts, Reading, UK.
[a] Peng Xian and Philip J. Klotzbach are joint-lead coauthors.
*Correspondence to:* Peng Xian (peng.xian@nrlmry.navy.mil)

## Abstract

Previous studies have noted a relationship between African dust and Atlantic tropical cyclone (TC) activity. However, due to the limitations of past dust analyses, the strength of this relationship remains uncertain. The emergence of aerosol reanalyses, including the Navy Aerosol Analysis and Prediction System (NAAPS) Aerosol Optical Depth (AOD) reanalysis, NASA Modern-Era Retrospective analysis for Research and Applications, Version-2 (MERRA-2) and ECMWF Copernicus Atmosphere Monitoring Service reanalysis (CAMSRA) enable an investigation of the relationship between African dust and

TC activity over the tropical Atlantic and Caribbean in a consistent temporal and spatial manner for 2003-2018. Although June-July-August (JJA) 550 nm dust AOD (DAOD) from all three reanalysis products correlate significantly over the tropical Atlantic and Caribbean, the difference in DAOD magnitude between products can be as large as 60% over the Caribbean and 20% over the tropical North Atlantic. Based on the three individual reanalyses, we have created an aerosol multi-reanalysis-consensus (MRC). The MRC presents overall better root mean square error over the tropical Atlantic and Caribbean compared

to individual reanalyses when verified with ground-based AErosol RObotic NETwork (AERONET) AOD measurements. Each of the three individual reanalyses and the MRC have significant negative correlations between JJA Caribbean DAOD and seasonal Atlantic Accumulated Cyclone Energy (ACE), while the correlation between JJA tropical North Atlantic DAOD and seasonal ACE is weaker. Possible reasons for this regional difference are provided. A composite analysis of three high versus three low JJA Caribbean DAOD years reveals large differences in overall Atlantic TC activity. We also show that JJA

Caribbean DAOD is significantly correlated with large-scale fields associated with variability in interannual Atlantic TC activity including zonal wind shear, mid-level moisture and SST, as well as ENSO and the Atlantic Meridional Mode (AMM), implying confounding effects of these factors on the dust-TC relationship. Further analysis indicates that seasonal Atlantic DAOD and the AMM, the leading mode of coupled Atlantic variability, are inversely related and intertwined in the dust-TC relationship.



## 1. Introduction

Saharan dust particles can affect weather and climate through both direct and indirect radiative and cloud processes, notably in association with boreal summer Saharan Air Layer (SAL) outbreaks. The SAL is a layer of hot and dry air that forms over continental West Africa and is then advected over the low-level moist marine boundary layer of the tropical Atlantic (Carlson
& Prospero, 1972). The SAL is often associated with the African Easterly Jet (AEJ) that can enhance vertical wind shear. Despite numerous observational and modeling studies that have examined the relationships between these aspects of the SAL and Atlantic TC activity, there are conflicting findings as to whether dust acts to generally inhibit or enhance tropical cyclogenesis and intensification. Some studies suggest negative impacts of the SAL's dust-laden dry air and the AEJ on TC activity (e.g., Dunion and Velden 2004; Lau and Kim 2007; Jones et al. 2007; Sun et al. 2008; Pratt and Evans 2008) while
others have focused exclusively on the dust particles themselves and have found a negative influence on TCs (e.g., Evan et al. 2006a; Rosenfield et al. 2007; Strong et al. 2018; Reed et al. 2019). Other studies have suggested little impact of the SAL on TCs (e.g., Braun 2010; Sippel et al. 2011; Braun et al. 2013), while others have posited a positive impact of dust on TCs through cloud-microphysical processes (e.g., Jenkins et al. 2008). Finally, others have suggested that there are contrasting influences through different mechanisms and for different TCs (Karyampudi and Pierce, 2002; Bretl et al. 2015; Pan et al.,
2018), highlighting the complexity of the dust-TC interaction.

African dust impacts the North Atlantic throughout the year, with its summer peak season (May-August) overlapping and leading the peak of the Atlantic hurricane season (August-October; Figure S1). As African dust outbreaks during the summer are often associated with the SAL, airborne dust has often been used as an indicator for the SAL (Dunion & Velden, 2004; Dunion, 2011; Tsamalis et al., 2013), although early season cases where the majority of the dust existed in the marine boundary
layer below the trade wind inversion instead of staying aloft were also found (Reid et al., 2003). Saharan dust and the SAL are frequently observed throughout the Caribbean and as far west as Central America and the North American continent during the boreal summer (e.g., Prospero, 1999; Reid et al., 2003; Dunion & Velden 2004; Nowottnick et al., 2011; Kuciauskas et al., 2018). Airborne dust associated with the SAL often extends to 5.5 km (500 hPa) off of western Africa, and becomes thinner as its top lowers and its base rises as it is advected westward, shrinking to below 2 km in the Caribbean and in the Gulf of
Mexico (Tsamalis et al., 2013). In some strong SAL cases, however, the top of the dust layer can reach 6 km (Reid et al., 2003; Colarco et al., 2003). During their trans-Atlantic transport, dust aerosols are from time to time observed to interact with TCs, as seen in satellite imagery (Figure 1).

African dust and its associated SAL has been hypothesized to impact TCs through a variety of mechanisms. Through scattering and absorbing sunlight, dust reduces solar radiation reaching the surface, thus cooling SSTs (e.g., Miller and Tegen, 1998; Lau
and Kim 2007; Evan et al., 2009). Colder SSTs provide TCs with less energy to initiate, develop, and maintain strength.



Through additional radiative heating of the dusty layer, mineral dust is also suggested to impact the structure, location and energetics of the AEJ (Tompkins et al., 2005; Wilcox et al., 2010; Reale et al., 2011) and African easterly wave (AEW) activity (Karyampudi and Carlson, 1988; Reale et al., 2009; Nathan et al., 2017; Jones et al., 2004; Ma et al., 2012; Grogan et al., 2016, 2017; Bercos-Hickey et al., 2017), thus having implications for tropical cyclogenesis. From a thermodynamic point of view,
Dunion and Velden (2004) have proposed that the dust-carrying SAL outbreaks could inhibit TC formation and development in the North Atlantic through three primary mechanisms, including dry air intrusion into the storm, enhancement of the local vertical wind shear associated with the enhanced AEJ, and stabilization of the environment due to radiative heating of the dust layer above the marine boundary layer.

Dust particles can also act as cloud condensation nuclei (Twohy et al., 2009; Karydis et al., 2011) and ice nuclei (DeMott et
al., 2003; Sassen et al., 2003) and affect cloud microphysics, weakening or strengthening convection depending on the environment (Khain, 2009). Focusing specifically on TCs, there is not a consistent conclusion among studies on whether the microphysical impacts of dust weaken or strengthen TCs (Jenkins et al., 2008; Rosenfeld et al., 2007; Zhang et al., 2007, 2009; Herbener et al., 2014; Nowottnick et al., 2018).

While dust aerosols can affect TC formation and development through radiative and cloud-microphysical impacts, TCs can in
turn impact dust aerosol spatial distributions through wet removal and dynamic flow (Herbener et al., 2016). AEWs, serving as seeding disturbances for TCs (Landsea, 1993), are shown to contribute to dust emission and transport (e.g., Westphal et al., 1987; Jones et al., 2003; Knippertz and Todd, 2010). Climate variability that affects TC activity can also impact African dust emission and transport over the North Atlantic and Caribbean. For example, ENSO was found to affect the emission and transport of African dust as well (Prospero & Lamb, 2003; DeFlorio et al., 2016), especially during the boreal winter (Prospero
& Nees, 1986; Evan et al., 2006b).

How all of these factors interact in the complex climate system and to what extent they can impact TC formation and intensification is still largely unknown. The goal of this study is to explore how the integrated interactions manifest themselves in the relationship between Saharan dust and Atlantic TC activity on seasonal to interannual time scales using state-of-the-art aerosol reanalysis data. This serves as a first step towards further understanding the dust-TC relationship and evaluating the
relative importance of different mechanisms. Previous empirical studies on the relationship between African dust and Atlantic TC activity are limited by uneven spatial and temporal sampling by satellite and in situ-based observations. The emergence of several aerosol reanalysis datasets, including the Navy Aerosol Analysis and Prediction System (NAAPS) Aerosol Optical Depth (AOD) reanalysis (NAAPS-RA, Lynch et al., 2016), the Modern-Era Retrospective analysis for Research and Applications, Version 2 (MERRA-2) aerosol reanalysis (Randles et al., 2017) and the Copernicus Atmosphere Monitoring
Service ReAnalysis (CAMSRA) (Inness et al., 2019) allow us to investigate this relationship in a more consistent manner over the acquired time periods to provide a degree of statistical robustness.



The Atlantic Main Development Region (MDR) (e.g., Goldenberg et al., 2001), including the Caribbean (10-20°N, 85-60°W) and the tropical North Atlantic (10-20°N, 60-20°W), is the focus region for this study (see also Figure 2 for a spatial representation of the two subregions). Statistical relationships between dust AOD (DAOD) and TC activity over the MDR are investigated using the three aerosol reanalyses and a multi-reanalysis-consensus (MRC) using the average of the three reanalyses. The results obtained herein also help us assess the potential of using DAOD to aid in future Atlantic seasonal hurricane forecasts.

In section 2, an introduction to the aerosol and large-scale environmental data and the analysis method is provided. Section 3 presents the DAOD climatology, its interannual variability over the MDR, and comparisons of the three aerosol reanalyses. This section also evaluates correlations between DAOD and Atlantic TC activity, as well as the relationship between DAOD and large-scale environmental conditions and climate modes. The sensitivity of the results to the definition of the regions, the number of composite years used, and the definition of dust seasons are provided in section 4. A discussion and conclusions are given in section 5.

## 2. Data and Methods

### 2.1 Aerosol data

A combination of aerosol reanalyses are used to describe the aerosol environment over the tropical North Atlantic and Caribbean. An aerosol multi-reanalysis-consensus (MRC) based on three aerosol reanalysis products, including the NAAPS-RA (Lynch et al., 2016) from US Naval Research Laboratory (NRL), MERRA-2 (Randles et al., 2017) from NASA, and CAMSRA (Inness et al., 2019) from ECMWF, is also generated and used. The analysis period is focused on 2003-2018, when all three aerosol reanalyses are available and both Terra and Aqua Moderate Resolution Imaging Spectroradiometer (MODIS) AOD retrievals are assimilated therein.

### 2.1.1 NAAPS AOD reanalysis

The NAAPS-RA product provides 550 nm speciated AOD at a global scale with 1°x1° degree spatial and 6-hourly temporal resolution for the years 2003-2018 (Lynch et al., 2016). This reanalysis uses a modified version of NAAPS and assimilates quality-controlled AOD retrievals from MODIS on Terra and Aqua and the Multi-angle Imaging SpectroRadiometer (MISR) on Terra (Zhang et al., 2006; Hyer et al., 2011; Shi et al., 2011). NAAPS characterizes anthropogenic and biogenic fine aerosol species (ABF), dust, sea salt and biomass burning smoke aerosols. The aerosol source functions were tuned regionally so that a best match between the model coarse and fine mode AODs and the Aerosol Robotic Network (AERONET) AODs can be obtained. Other model processes, e.g. deposition, were also tuned to minimize the AOD difference between the model and quality-controlled satellite AOD retrievals. NOAA Climate Prediction Center (CPC) MORPHing (CMORPH) precipitation derived from satellite observations (Joyce et al., 2004) is used to correct precipitation biases in the tropics for improved AOD





analyses through wet deposition processes (Xian et al., 2009). The reanalysis captures the decadal AOD trends detected using standalone satellite products in other studies (e.g., Hsu et al., 2012; Zhang et al., 2017), demonstrating the quality of reanalysis products for climate studies. The NAAPS-RA data for May 2017 - November 2018 was generated by assimilating MODIS DA-quality AOD only without MISR AOD assimilation because of the unavailability of MISR DA-quality data at the time of this study. The impact of not including MISR is expected to be minor as MISR provides only about 10% of the total assimilated AOD data. Additionally, differences between monthly mean DA-quality AOD over the MDR region derived using both MODIS and MISR versus using only MODIS were found to be negligible (not shown).

### 2.1.2    MERRA-2 AOD reanalysis

As part of the upgrade from the original MERRA reanalysis (Rienecker et al., 2011) based on the Goddard Earth Observing System (GEOS) Earth system model, MERRA-2 now incorporates assimilation of AOD from a variety of remote sensing sources, including AERONET, MODIS, and MISR after 2000, and AVHRR before 2002. The aerosol module used for MERRA-2 is the Goddard Chemistry, Aerosol, Radiation, and Transport model (GOCART; Chin et al. 2000; Colarco et al., 2010), which provides simulations of dust, sea salt, black and organic carbon, and sulfate aerosols, and is run radiatively coupled to the GEOS AGCM. A detailed description and validation of the AOD reanalysis product can be found in Randles et al. (2017) and Buchard et al. (2017). For the purpose of this study, monthly mean DAOD at 550 nm with 0.5° latitude and 0.625° longitude spatial resolution is used.  MERRA-2's longer data record (1981-present) would have made it an ideal candidate for a longer-period analysis of the relationship between DAOD and TCs. However, the volcanic eruptions of Pinatubo (1991) and El Chichon (1982) result in high AOD for several years following each event, and the particle property assumptions in MERRA-2 do not properly apportion the assimilated AOD increments among the simulated aerosol species. For the 2003-2018 time period, MERRA-2 AOD data has similar validation statistics compared to NAAPS-RA and CAMSRA at the sites located off of the coast of West Africa and the Caribbean, as shown in Table 1.

### 2.1.3 CAMSRA AOD reanalysis

Under the banner of the Copernicus Atmosphere Monitoring Service (CAMS), operated by ECMWF on behalf of the European Commission, a new global reanalysis of atmospheric composition has been produced: CAMSRA (Inness et al., 2019). This is the successor to the MACC reanalysis (Inness et al., 2013) and CAMS interim reanalysis (Flemming et al., 2017) produced previously at ECMWF. The dataset spans the period 2003–2018 and is being continued for subsequent years. The model component is based on the same Integrated Forecasting System (IFS) used at ECMWF for weather forecasting and meteorological reanalysis, but at a coarser resolution and with additional modules activated for prognostic aerosol species (dust, sea salt, organic matter, black carbon and sulphate) and trace gases. The impact of the aerosols (and ozone) on radiation and thereby on meteorology is included in the model. For aerosols, observations of total AOD at 550nm are assimilated from MODIS (Terra and Aqua) for the whole period, and from AATSR for 2003–2012, using a 4D variational data assimilation





system with a 12-hour data assimilation window along with meteorological and trace gas observations. The speciated AOD products used in this study are available at a 3-hourly temporal resolution and a ~0.7° spatial resolution. Model development

has generally improved the speciation of aerosols compared with earlier reanalyses, and evaluation against AERONET is largely consistent over the period of the reanalysis. There is a known issue regarding a significant overestimation of sulfate near outgassing volcanoes; however this is unlikely to have much relevance to the regions considered in this study.

### 2.1.4 AOD multi-reanalysis-consensus (MRC)

Based on the three aerosol reanalysis products described above, we made a MRC product following the multi-model-ensemble

method of the International Cooperative for Aerosol Prediction (ICAP, Sessions et al., 2015; Xian et al., 2019). The MRC is a consensus mean of the three individual reanalyses, with a 1°x1° degree spatial and monthly temporal resolution. Speciated AODs and total AOD at 550nm for 2003-2018 are available. This new product is validated with ground-based AERONET observations for African-dust-influenced regions, including the western coast of North Africa and the Caribbean Sea. Validation results in terms of RMSE for total and coarse-mode AODs are presented in Table 1. Similar to the ICAP multi-

model-ensemble evaluation result, the MRC is found to generally be the top performer among all of the reanalyses for the study region.

### 2.1.5 AErosol RObotic NETwork (AERONET) fine and coarse mode AOD

AERONET is a ground-based global scale sun photometer network that includes instruments to measure sun and sky radiance at wavelengths ranging from the near ultraviolet to the near infrared during daytime hours. This network has been providing

high-accuracy and high-quality measurements of aerosol properties since the 1990s (Holben et al., 1998; Holben et al., 2001) and is often used as the primary dataset for validating aerosol optical properties in satellite retrievals and model simulations (e.g., Levy et al., 2010; Colarco et al., 2010; Kahn & Gaitley, 2015). Only cloud-screened, quality-assured version 3 Level 2 AERONET data are utilized in this study (Giles et al., 2019). AERONET multiple wavelengths measurement were used to derive both fine and coarse mode AODs at 550 nm based on the Spectral Deconvolution Method (SDA) of O'Neill et al.

(2003). The SDA product was verified with in situ measurements (Kaku et al., 2014) and was shown to be able to capture the full modal properties of fine and coarse particles. Temporally, AERONET data are averaged into 6-hr bins centered at the regular model output times of 0, 6, 12 and 18 UTC. Monthly mean AERONET AOD is derived only when the total number of 6-hr AERONET data is greater than 18 to ensure temporal representativeness.

### 2.2 Tropical Cyclone data – HURDAT2

Atlantic basin TC data were taken from the Atlantic hurricane database version 2 (HURDAT2; Landsea & Franklin, 2013). This dataset contains six-hourly information (including position, maximum sustained winds, and central pressure – where available) for every TC observed in the Atlantic basin dating back to 1851.



## 2.3 Atmospheric data - ERA-Interim Reanalysis

The ERA-Interim Reanalysis (Dee et al., 2011) is a global atmospheric reanalysis produced by the European Centre for Medium–Range Weather Forecasts (ECMWF) that uses a 4-dimensional variational analysis with a 12-hour analysis window. The spectral resolution of this data is approximately 80 km (T255) and is available on 60 vertical levels from the surface to 0.1 hPa and is available from January 1979 – August 2019. The monthly mean large-scale fields, including vector wind, atmospheric temperature and relative humidity data on pressure levels are used.

## 2.4 Oceanic data - NOAA OI SST

The National Oceanic and Atmospheric Administration (NOAA) Optimum Interpolation (OI) SST product (Reynolds et al., 2002) is utilized for SST calculations. NOAA OI SST v2 uses a combination of in-situ data, satellite data, SSTs simulated by sea-ice cover, and bias adjustments to arrive at its final estimate of SSTs. NOAA OI SST V2 data is available on a 1° x 1° grid from November 1981-present.

## 2.5 Climate indices

The Oceanic Nino Index (ONI), defined to be a three-month average of the Niño 3.4 (5°S-5°N, 170-120°W) index (Barnston et al., 1997) based on centered 30-year periods which are updated every five years, is utilized to represent the state of El Niño-Southern Oscillation (ENSO). This index is also used by NOAA to identify ENSO events.

The SST component of the AMM (Kossin & Vimont, 2007) is investigated to assess the relationship between DAOD and tropical Atlantic oceanic conditions. While the index is not standardized, we have standardized it by its 1981-2010 average and standard deviation.

## 2.6 Derived tropical cyclone indices

The genesis potential index (GPI) was calculated using monthly-averaged ERA-Interim data following Emanuel and Nolan (2004). The maximum potential intensity (MPI) was calculated using monthly-averaged ERA-Interim temperature and moisture and NOAA OI SST following Bister and Emanuel (2002).

## 2.7 Statistical correlation calculations and significance tests

The correlations between variables of interest are based on the Pearson correlation coefficient. Statistical significance is assessed at the 95% level using a two-tailed Student's t-test. Correlations $>= 0.51$ are statistically significant given that a 16-year time period (e.g., 2003-2018) is investigated here. For partial correlation analysis, partial correlations $>=0.55$ are statistically significant at the 95% level with 13 degrees of freedom. The criteria for statistical significance with various degrees of freedom can also be obtained at: https://www.esrl.noaa.gov/psd/data/correlation/significance.html.



## 3. Results

### 3.1 Dust aerosol optical depth over the MDR (2003-2018)

Figure 2 shows the MRC monthly DAOD climatology based on the 2003-2018 average as well as the ratio of DAOD to total AOD for June-October over the tropical Atlantic. Climatologically from June-October, the majority of airborne dust originates
from the Sahara Desert, in contrast to the winter season when a significant amount of dust is emitted over the Sahel and southern Sahara (Engelstaedter & Washington, 2007). This dust is then transported westward over the Atlantic and eventually to the Caribbean, largely within the 10-25°N latitude belt. The transported African dust covers most of the Atlantic hurricane MDR, which spans the tropical North Atlantic and Caribbean. The DAOD over the Atlantic is, on average, much higher in June, July, and August (JJA) than in September and October because of higher emissions over the African continent in the
former months (Carlson and Prospero 1972; Engelstaedter and Washington, 2007; Dunion & Marron, 2008; Dunion, 2011). DAOD is also much higher over the tropical Atlantic than over the Caribbean, as dust aerosols are removed by wet and dry processes during long-range transport. The MRC shows that dust aerosols are the dominant contributor to the total AOD in the MDR during most of the hurricane season (June-October). The DAOD accounts for about 50-60% of the total AOD over the tropical North Atlantic and around 30-50% over the Caribbean for JJA. This suggests that the total AOD can be a relatively
good indicator of DAOD in the tropical North Atlantic but is not as good of an indicator in the Caribbean for JJA. The DAOD contribution to total AOD is about 10-20% less for September and October. Considering the potential larger forcing by airborne dust in JJA than in September and October, the focus season in this study is JJA.

As transport of Saharan dust across the Atlantic during summer is often associated with SAL outbreaks, which are approximately centered around 700 hPa (Dunion & Velden, 2004; Dunion, 2011), monthly climatological 700 hPa relative
humidity (RH) and horizontal wind are also shown in Figure 2. Climatologically, the MDR is dominated by the mid-level easterly jet (7-8 m s$^{-1}$) during JJA and weaker easterlies (5-6 m s$^{-1}$) during September and October. Over the Caribbean, the wind direction veers slightly towards the north for all of the studied months and relates to this region being typically positioned on the west side of the climatological Atlantic subtropical ridge. 700 hPa RH is on the order of 40% and 50% for the tropical Atlantic and the Caribbean respectively for JJA and is about 10% higher in September and October. RH is higher in the
Caribbean than in the tropical North Atlantic as the impact of dry air from the SAL and from upper-level subsidence becomes weaker from east to west. For context, the Atlantic moist tropical sounding, defined in Dunion (2011), has average 700 hPa winds of 3.6 m s$^{-1}$ at 112° (wind direction) and 66% RH, while the mean SAL sounding has corresponding values of 7.8 m s$^{-1}$ at 91° and 34% RH.

Figure 3 shows the monthly mean AERONET version 3 L2 and MRC 550 nm modal AOD time series at four AERONET sites
that are primarily influenced by African dust. From east to west, these sites include Dakar, Senegal (14.4°N, 17.0°W), Cape Verde (16.7°N, 22.9°W), Ragged Point, Barbados (13.2°N, 59.4°W), and La Parguera, Puerto Rico (18.0°N, 67.0°W), which



are also marked in Figure 1. The boreal summer peak dust activity (i.e., JJA) is highlighted. Dust aerosols are typically considered coarse-mode, although there may be a very small amount of mass in fine-mode. The fine-mode AOD observed in AERONET measurements for these sites are normally dominated by pollution and biomass-burning smoke. Dakar and Cape

Verde experience dust aerosols throughout the year, with a peak in JJA and a weaker secondary peak during the boreal winter, which is associated with dust emissions from the Sahel. The Ragged Point and La Parguera sites, which are remote receptor sites in the Caribbean, are influenced by African dust predominantly during boreal summer, thus displaying a pronounced peak of total and coarse-mode AODs in JJA. The secondary AOD peak during winter at Dakar and Cape Verde is generally not observed at Ragged Point or La Parguera, as African dust is transported to the south following the low-to-mid-level trade wind

flow, occasionally reaching South America (Prospero, 2014).

Figure 3 also shows the bias, the root mean square error (RMSE) of MRC and the correlation ($r$) between MRC and AERONET for monthly AODs at each of the four AERONET sites. Overall, MRC follows the AERONET interannual and seasonal variability for the total AOD quite well, and to a slightly lesser extent for the coarse-mode AOD. Coarse-mode aerosols include dust and sea salt, but are dominated by dust aerosols over the tropical North Atlantic for JJA (Figure 2c, speciated AOD cannot

be obtained from AERONET measurements). The correlation is >~0.9 for the coarse-mode AOD and tends to be better at the long-range transport sites (i.e., Ragged Point and La Parguera) than the sites close to the dust source (i.e., Dakar and Cape Verde). The correlation is slightly lower in the source area than in the long-range transport region because there are large uncertainties in emissions and strong gradients due to local aerosol sources in the source area. Also, the source area allows less time for AOD data assimilation to correct aerosol mass loads, in addition to a lower signal/noise ratio of AOD retrievals over

land than over water (e.g., Levy et al., 2005). The small bias, low RMSE and the high correlation with AERONET data illustrate the ability of MRC to capture the aerosol environment in the MDR. In addition, all of the three individual aerosol reanalyses that form the foundation of MRC have similar verification scores against AERONET, though the MRC typically has a better verification score than the individual reanalyses (Table 1). If we were to give different qualitative ratings for the three individual reanalyses for the study area, MERRA-2 is slightly better over North Africa and NAAPS-RA is slightly better

over the Caribbean in terms of coarse-mode and total AOD RMSEs.

Figure 4 shows the time series of monthly mean and regionally-averaged DAOD from MRC and the three contributing reanalyses from 2003-2018 for the tropical North Atlantic and Caribbean as defined in Fig. 2. The DAODs from the three reanalyses have similar seasonal and interannual variability and are highly correlated, with $r >= 0.95$ for the entire 16-yr period and $r >= 0.85$ for JJA for both regions based on monthly means. The magnitudes of JJA DAOD from the three individual

reanalyses are comparable over the tropical North Atlantic, with a 20% maximum difference among the three products based on JJA DAOD. The climatological average of JJA DAOD over the tropical North Atlantic is 0.21. The difference can be as much as ~0.06 (a ~60% difference between member products) for the Caribbean. The climatological average of JJA DAOD in the Caribbean is 0.10. Since the total and coarse-mode AOD verification statistics for the three products at the Caribbean AERONET sites are similar (Table 1), the DAOD difference is most likely due to the different partitioning of aerosol species



(e.g., dust versus sea salt aerosols) during the total AOD data assimilation process. This is related to the fact that total AOD is the only aerosol property constrained by satellite observations through AOD data assimilation in all three aerosol reanalysis products, while speciated AOD is not constrained (Lynch et al., 2016; Randles et al., 2017; Inness et al., 2019). Nevertheless, the DAOD from the MRC is likely the most reliable given the generally better performance of multi-aerosol model consensus compared to individual aerosol models (Sessions et al., 2015; Xian et al., 2019).

### 3.2 Relationship between North Atlantic TC activity and JJA DAOD


Accumulated Cyclone Energy (ACE) is often utilized to represent TC activity and is defined to be the square of the one-minute maximum sustained wind speed at each six-hourly interval when a tropical or subtropical cyclone (with maximum sustained winds >=34 kt) is present (Bell et al., 2000). Basin-wide ACE is used here, as it is assumed that MDR conditions affect to some extent, the ACE of all storms that pass through the MDR, including those that later moved out of the MDR. For example,

we hypothesize that in an active dust year, the suppressed conditions in the MDR would make for weaker, less organized AEWs that have less of a chance for formation even if they do eventually move out of the MDR. Later we show in Table 2 that using ACE generated in the Caribbean domain yields a consistent (same sign) yet stronger correlation relationship between DAOD and ACE.

Atlantic TC activity shows a statistically significant relationship with regionally-averaged Caribbean JJA DAOD. Figure 5a

displays this relationship, with higher Caribbean DAOD correlating ($r = -0.61$ with MRC DAOD and $r$ of similar magnitudes from all three individual reanalyses and exceeding the two-tailed 95% statistical significance level) with quieter Atlantic hurricane seasons as quantified by ACE. While tropical North Atlantic DAOD and Caribbean DAOD in JJA correlate strongly ($r = 0.88$), Figure 5b shows that the relationship between DAOD and ACE is weaker in the tropical North Atlantic than in the Caribbean. The correlation between tropical North Atlantic DAOD and ACE is -0.41, which falls below the statistical

significance threshold (correlations with MERRA-2 and CAMSRA DAODs are also insignificant). We will show in the next section that the relationship between large-scale fields known to impact Atlantic TC activity also tend to have higher correlations with JJA Caribbean DAOD than with JJA tropical North Atlantic DAOD.

Given the strength of the relationship between Caribbean DAOD and seasonal Atlantic ACE, we next investigate the relationship in extreme JJA DAOD seasons. We take the three seasons from 2003-2018 when JJA Caribbean DAOD was at

its highest levels and when it was at its lowest levels.

The three seasons with the highest JJA Caribbean DAOD were 2018, 2015 and 2014 in descending order, and the three seasons with the lowest JJA Caribbean DAOD were 2005, 2011 and 2017 in ascending order based on MRC. The left column of Figure 6 shows DAOD composites for the three high and the three low JJA Caribbean DAOD seasons and their differences. DAOD is not only higher over the MDR in the high Caribbean DAOD seasons, but dust aerosols are transported farther to the west.

DAOD differences between the extreme high and low DAOD seasons over the tropical North Atlantic and Caribbean are



~0.05-0.08. In fact, the three high dust years have roughly 60% more DAOD in the Caribbean than the three low dust years (regional average of 0.13 vs. 0.08), while these differences are not as large in the tropical North Atlantic. The transport pathway of dust is also shifted slightly to the south in the tropical North Atlantic in the three high DAOD seasons.

The right column of Figure 6 displays the JJA-averaged 850 hPa winds and 700 hPa RH for the three high and three low JJA Caribbean DAOD seasons, and the difference between these high and low seasons. Large-scale conditions over the Caribbean during JJA were much less conducive for TCs in the high DAOD seasons, with drier middle levels (>10% relative humidity difference) and stronger easterly trade winds (2-4 m s$^{-1}$ stronger), implying an overall less hurricane-favorable dynamic and thermodynamic environment. This is consistent with the depiction of the thermodynamic structure of the SAL by Dunion and Velden (2004). The stronger AEJ associated with higher DAOD is also consistent with modeling studies when aerosol radiative effects are taken into account (Tompkins et al., 2005). Associated with high Caribbean DAOD, the position of the center of the Azores High was slightly shifted to the southwest, which facilitates stronger dust transport into the Caribbean. This is in agreement with the findings of Riemer and Doherty (2006) who suggested the importance of the position of the Azores High in African dust transport across the Atlantic, although their study was only during the boreal winter. As would be expected from these large-scale conditions, Atlantic TC activity was much higher in low JJA Caribbean DAOD seasons.

Table 2 displays observed Atlantic TC activity as well as the ratios of observed average seasonal Atlantic TC activity in the three low JJA Caribbean DAOD seasons versus the three high Caribbean DAOD seasons. Atlantic basin-wide numbers of tropical depressions, named storms, hurricanes, major hurricanes and ACE are higher by a factor of ~1.9 - ~2.8 in the three low than in the three high Caribbean DAOD seasons. The ratios are even higher (e.g., 12 times higher for ACE) for TC activity in the Caribbean. The 2018 Atlantic hurricane season was an interesting case, as it was an above-average overall hurricane season (as measured by ACE), but much of the ACE that was generated that year occurred outside of the tropics (>23.5°N) (Saunders et al. 2020). Very little activity occurred in the Caribbean in 2018.

In addition, the ratio for major hurricanes is higher than the ratios for tropical depressions, named storms, and hurricanes, indicating a stronger relationship between dust aerosols and intense storms than between dust aerosols and weak storms. A total of 17 major hurricanes were observed in the Atlantic in the three low Caribbean DAOD seasons, compared with only 6 major hurricanes in the three high Caribbean DAOD seasons. The three low Caribbean DAOD seasons had six continental United States major hurricane landfalls (2005 Hurricanes Dennis, Katrina, Rita, and Wilma and 2017 Hurricanes Harvey and Irma), while the three high Caribbean DAOD seasons had one continental United States major hurricane landfall (2018 Hurricane Michael).

Figure 7 displays the named storm formation location of all Atlantic TCs in the three seasons with the highest JJA Caribbean DAOD and the three seasons with the lowest Caribbean DAOD , along with the maximum intensity that these TCs reached. As would be expected from the differences in large-scale conditions noted earlier, TCs that became major hurricanes formed


much more frequently south of 20°N in the three lowest Caribbean DAOD seasons than in the three highest Caribbean DAOD seasons. These differences were most pronounced in the Caribbean, with only one named storm (Hanna in 2014) forming in the Caribbean in the three highest JJA Caribbean DAOD seasons. In the three lowest Caribbean JJA DAOD seasons, 11 named

storms formed in the Caribbean.

### 3.3 Relationship between JJA DAOD and large-scale atmosphere/ocean fields

We next examine the relationship between JJA DAOD and large-scale atmosphere/ocean fields. In this analysis, we begin by focusing on several fields that have been documented in prior research to significantly impact Atlantic TC activity: 850 hPa zonal wind (850 hPa U), 200 hPa zonal wind (200 hPa U), zonal wind shear between 200 hPa and 850 hPa, 700 hPa RH, 850

hPa relative vorticity and SST (Gray, 1968; Saunders et al., 2017). More active Atlantic hurricane seasons are typically associated with anomalous westerly 850 hPa U (e.g., weaker trade winds), anomalous easterly 200 hPa U (counteracting prevailing upper-level westerlies) and thus weaker wind shear, higher 850 hPa relative vorticity, higher mid-level relative humidity and anomalously warm SSTs. We investigate the relationships with DAOD in the tropical North Atlantic and the Caribbean as defined in Figure 2.

Figure 8 displays the correlation between regionally-averaged MRC JJA DAOD in the Caribbean and the six large-scale fields just discussed. Higher JJA Caribbean DAOD is associated with stronger 850 hPa easterly trades and increased 200 hPa upper-level westerlies (and hence stronger vertical wind shear), drier air at 700 hPa and anomalously cool SST across the MDR. Weaker 850 hPa relative vorticity also predominates over most of the Caribbean. However, almost no correlation is found between Caribbean JJA DAOD and 850 hPa relative vorticity in the tropical North Atlantic, possibly due to the counteracting

role between the covariability of African dust emissions and AEWs - as was inferred from a positive correlation between the two by Karyampudi and Carlson (1988). This result is also consistent with Figure 7 which shows more named storms and therefore likely stronger AEW activity right off of the coast of west Africa between 10-20°N in high dust DAOD years. This result reflects the complexity of the TC-dust interaction relationship. In addition, the vorticity field is extremely noisy. Figure 8f is extended to include the eastern and central tropical Pacific in order to investigate the potential relationship between ENSO

and DAOD, with Caribbean DAOD showing a significant positive relationship with ENSO.

The relationship between the same six large-scale fields and JJA tropical North Atlantic DAOD is considerably weaker, with lower correlations observed for all six fields (Figures S2). In addition, the regions with significant correlations decrease in spatial extent relative to the Caribbean DAOD correlations shown in Figure 8.

We next investigate Maximum Potential Intensity (MPI), an integrated TC index, which combines a list of key factors (similar

to those explored above). MPI assesses how conducive atmospheric thermodynamic conditions are for TC intensification, providing a theoretical limit of the strength of a TC (Holland, 1997; Bister and Emanuel, 1998). Figures 9a and c show the correlations between the Caribbean region-averaged JJA DAOD and JJA/ASO MPI calculated based on Bister and Emanuel





(1998). Consistent with the results for the individual large-scale fields, JJA MPI over the MDR exhibits strong negative correlations with JJA Caribbean DAOD (~ -0.7, also Table 3). One of the primary inputs to the MPI calculation is SST, so the

strong inverse relationship between MPI and DAOD is expected given the strong inverse relationship between SST and DAOD. The correlation is significant but weaker for ASO MPI for most of the MDR, except for the eastern tropical North Atlantic, where the negative correlation drops below statistical significance. The negative correlation of JJA/ASO MPI with JJA tropical North Atlantic region-averaged DAOD is much weaker than that with JJA Caribbean DAOD in the MDR (Figure 9b and d) and drops below statistical significance in the Caribbean.

The genesis potential index (GPI) is another integrated TC index that is often used to provide an estimate of the potential for tropical cyclogenesis (e.g., Emanuel and Nolan, 2004; Camargo et al., 2007). Monthly GPI is calculated following Emanuel and Nolan (2004). Figure 10 shows the correlation between region-averaged JJA DAODs and JJA and ASO GPI. Consistent with the results for the individual large-scale fields and MPI, JJA GPI over the MDR exhibits strong negative correlations (~- 0.7, also Table 3) with JJA Caribbean DAOD. Similar to MPI, GPI is also directly related to SST via the potential intensity

term, so given the negative correlation between DAOD and SST, we would expect a negative correlation between GPI and DAOD. Other terms also comprise the GPI, including vertical wind shear and mid-level moisture, which also correlate negatively with DAOD. The correlation remains statistically significant but is weaker for ASO GPI for most of the MDR. The exception is the eastern tropical North Atlantic, where the negative correlation drops below 95% statistical significance. The negative correlation of JJA/ASO GPI with JJA tropical North Atlantic region-averaged DAOD is much weaker than that

with JJA Caribbean DAOD in the MDR (Figure 10b and d), which is also consistent with the result for the individual large-scale fields and MPI.

Table 3 summarizes the relationship between large-scale atmosphere/ocean fields, MPI, GPI, and DAOD, with correlations displayed between JJA region-averaged DAOD and concurrent region-averaged fields (i.e., JJA-averaged), as well as the large-scale region-averaged fields during the peak of the Atlantic hurricane season from August-October. 850 hPa relative vorticity

is excluded as no statistically significant correlations are found. While the correlations between the other five large-scale fields, MPI, GPI, and DAOD tend to weaken from JJA to ASO, the correlations remain significant for all of these large-scale fields, and the integrated TC indices, in the Caribbean during ASO. For the tropical North Atlantic, JJA DAOD has much weaker and insignificant correlations with JJA 200 hPa zonal wind, wind shear and 700 hPa RH compared to those for the Caribbean. However its negative correlation with SST is as strong as that for the Caribbean in JJA and remains statistically significant

from JJA to ASO, although the magnitude of the correlation is weaker in ASO. These contribute to a negative correlation with MPI and GPI and a stronger correlation during JJA than during ASO.

Part of the reason for the rapid decrease in the strength of the correlations in the tropical North Atlantic is due to relatively low correlations between JJA and ASO values of large-scale parameters in that portion of the basin, indicating a lack of persistence





in atmosphere/ocean conditions when compared with the Caribbean (Table 4). The persistence of the 700 hPa RH and 850
hPa U fields in the tropical North Atlantic is especially low compared to other fields.

## 3.4 Relationship between JJA DAOD and large-scale climate modes

We next explore the relationship between JJA-averaged DAOD and two large-scale climate modes that have been documented
in many studies to impact Atlantic TC activity: ENSO (e.g., Gray, 1984; Goldenberg & Shapiro, 1996; Klotzbach, 2011;
Klotzbach et al., 2018) and the AMM (e.g., Kossin & Vimont, 2007; Patricola et al., 2014). El Niño typically reduces Atlantic
TC activity through several mechanisms including increasing westerly wind shear especially over the Caribbean (Gray, 1984)
and through upper-tropospheric warming, causing increased static stability and inhibiting deep convection (Tang & Neelin,
2004). The AMM has also been suggested in prior research to significantly impact Atlantic TC activity (Kossin & Vimont,
2007), especially when combined with ENSO (Patricola et al., 2014). A positive phase of the AMM is associated with a warmer
than normal tropical Atlantic, anomalously low sea level pressure and anomalously weak trade winds - all of which favor
Atlantic TC formation (Kossin & Vimont, 2007).

Table 5 displays the correlations between JJA regionally-averaged DAODs from the different aerosol reanalysis products and
the JJA and ASO ENSO (as represented by the ONI) and AMM indices. There is a positive correlation between JJA Caribbean
DAOD and the concurrent (significant at the 90% level) and ASO ONI (significant at the 95% level) using MRC and NAAPS-
RA, and the correlations with the ONI increase from JJA to ASO. The correlation between JJA tropical North Atlantic DAOD
and JJA/ASO ENSO is not significant, however, consistent with previous studies (Lau and Kim, 2007; Doherty et al., 2014).
ENSO events climatologically intensify from boreal summer to boreal autumn (Harrison & Larkin, 1998), which may be part
of the reason for the increase in significance of the correlations from JJA to ASO.  In addition, this likely also explains part of
the reason why Atlantic TC activity correlates more strongly with Caribbean DAOD than with tropical Atlantic DAOD, given
the pronounced impact that ENSO has on the Caribbean large-scale environment (Gray, 1984). Figure S3, in which JJA
composites of MRC DAOD, 850 hPa horizontal wind and 700 hPa RH for the three top El Niño and La Niña years (based on
JJA ONI) are shown, corroborates that stronger dust transport into the Caribbean occurs during El Niño years without
necessarily having stronger emissions over Africa and high DAOD over the tropical North Atlantic. We also note that 2015 is
both an El Niño and high DAOD year, while 2011 is both a La Niña and low DAOD year. Removing the two overlapping
years in the composites (not shown) leads to similar results except that the DAOD is higher in La Niña years than in El Niño
years in the tropical North Atlantic, supporting the insignificant correlation between ENSO and tropical North Atlantic DAOD.

The correlations between JJA Caribbean DAOD and JJA AMM are consistently strong and negative (~-0.7) with all of the
aerosol reanalyses (Table 5). As might be expected given that the signal of the AMM is climatologically strongest in the boreal
spring and weakens in the boreal summer and fall (Kossin & Vimont, 2007), correlations of JJA DAOD are weaker with the
ASO AMM than with the JJA AMM.  Negative correlations are also obtained between the JJA tropical North Atlantic DAOD



and the JJA AMM, while the correlation with the ASO AMM is weak and not statistically significant. The negative correlations between JJA DAODs and the AMM are consistent with Evan et al. (2011)'s study on the radiative effect of dust aerosols on the AMM.  That study showed DAOD to be not only negatively correlated with the AMM but that its variability was found to excite the AMM on interannual to decadal time scales.

So far we have shown that Caribbean DAOD is correlated with Atlantic basin-wide ACE as well as two-large scale climate
modes, ENSO and the AMM, which have both been shown to also impact Atlantic TC activity. To remove the influence of these climate modes from the relationship between DAOD and ACE, we use partial correlation analysis (Sharma et al., 1978). Table 6 shows the partial correlation matrix between JJA Caribbean and tropical North Atlantic DAOD and annual Atlantic basin-wide ACE while controlling for the ONI and AMM indices, respectively. Removing the influence of ENSO causes little change in the negative correlation between MRC Caribbean JJA DAOD and ACE.  The correlation remains statistically
significant, suggesting that ENSO is not primarily responsible for the negative correlation between Caribbean DAOD and Atlantic ACE, at least during the study period. The correlation between tropical North Atlantic JJA DAOD and ACE also changes little, although the correlation is weak and not statistically significant initially. We note that the correlation between ONI and ACE is very weak and is not significant during the 2003-2018 study period, partially due to the extremely active 2004 Atlantic hurricane season which occurred despite a weak El Niño event.

In contrast to the findings of removing ENSO from the DAOD-ACE relationship, after removing the influence of the AMM, the correlation between Caribbean JJA DAOD and Atlantic ACE is much weaker and drops to insignificant levels, suggesting that the AMM is an important factor in the dust-TC relationship. However, it is hard to argue that the AMM is the determining factor in the dust-TC relationship, as the correlation of ACE with JJA Caribbean DAOD is slightly higher than with the JJA AMM (-0.61 vs. 0.59). When the partial correlation is calculated between the AMM and ACE while removing the Caribbean
dust (DAOD) influence, the correlation drops from 0.59 to an insignificant level ($r = \sim 0.26$) independent of season examined. This indicates that Caribbean DAOD is, in turn, an important factor in the AMM-TC relationship. When the TATL JJA DAOD is removed, the correlation between AMM and ACE is also reduced. It appears that the AMM/DAOD/TC relationship is strongly intertwined. This is physically feasible, as the dust radiative forcing can result in cooler SST, which can introduce anomalously high sea level pressure and stronger trade winds (and therefore stronger vertical wind shear), which is
characteristic of a negative AMM. This air-sea coupled response to dust radiative forcing can act over relatively short timescales (e.g., one to two months) (Evan et al. 2011). Additionally, the dryness of the dusty air and the radiative heating of the dust layer, can lead to stronger vertical wind shear and a more stable lower atmosphere, all in line with a negative AMM that creates an environment that is detrimental for TC formation and development. It could be argued that the negative correlation between DAOD and Atlantic TC activity may be a result of forcing of the AMM by African dust.

**4. Sensitivity tests**



The sensitivity of our result to the domain definitions of the tropical North Atlantic and the Caribbean regions is explored by defining equal areas (shifting the separation longitude to 52.5°E) for the two regions within the MDR, as well as by expanding the two regions by 5° latitude to the north (e.g., to 25°N). Neither of the two new definitions for regions significantly change the correlations between JJA regionally averaged DAOD and large-scale fields (Table S1, S2). The correlations between the

JJA Caribbean DAOD and ACE are -0.59/-0.58 for the longitudinal and latitudinal shift respectively. These correlations are only slightly lower than the -0.61 correlation using the default regional definitions. The correlations between the JJA TATL DAOD and ACE remain insignificant (Table S3).

While airborne dust can impact TC activity, once TCs form and develop, both precipitation and strong winds can significantly remove these dust particles. However, the removal effect by TCs cannot primarily explain the negative DAOD-TC relationship.

This is because peak dust activity occurs from June to August with larger DAOD in June and July than in August over the Atlantic, suggesting that peak DAOD in general leads the peak TC season by ~1-2 months (Figure S1). Using June-July average DAOD instead of June-August average DAOD changes our results only slightly. For example, the correlation between June-July Caribbean DAOD and Atlantic ACE is -0.58, only slightly lower than that using JJA DAOD ($r = -0.61$) (Table S3). This has implications for using DAOD as an indicator for seasonal TC forecasts which are often updated in early August.

The sensitivity of the composite analysis of high versus low JJA Caribbean DAOD years to the number of years is also explored by using two and four years for composites in addition to three years (Table S4). Consistent results are found across all sensitivity tests. Atlantic basin-wide numbers of tropical depressions, named storms, hurricanes and major hurricanes, and ACE are higher (by a factor of 1.6-5) in the low than in the high JJA Caribbean DAOD seasons. The ratios of observed average seasonal Atlantic hurricanes and major hurricanes in the low versus the high JJA Caribbean DAOD seasons are generally

higher than those of tropical depressions and named storms, suggesting a stronger correlation relationship between dust aerosols and intense storms than weak storms.

## 5. Conclusions and Discussions

The relationship between African dust and Atlantic tropical cyclone (TC) activity has been analyzed in many prior studies (e.g., Dunion & Velden 2004; Evan et al. 2006a; Braun et al. 2013; Pan et al. 2018). This study has revisited this relationship

with a statistical analysis using three newly available aerosol reanalyses, the Naval Aerosol Analysis and Prediction System reanalysis (NAAPS-RA), the Modern-Era Retrospective analysis for Research and Applications, Version 2 (MERRA-2) aerosol reanalysis, the Copernicus Atmosphere Monitoring Service ReAnalysis (CAMSRA), and a multi-reanalysis-consensus (MRC) based on the three reanalyses for the period 2003-2018. The datasets are validated with ground-based observations for modal (fine, coarse and total) aerosol optical depth (AOD). The MRC data is primarily used in this study as it generally has

better verification results than any of the individual reanalysis products. To our knowledge, this is the first climate study using a multi-reanalysis consensus to represent aerosol conditions. Our findings are summarized below:



1. Total AOD of the three aerosol reanalysis products are similar for the Atlantic Main Development Region, however, AOD attributed to individual aerosol species (such as dust aerosols) can be quite different among the three reanalysis products (Figure 4). June-July-August (JJA) dust AOD (DAOD) magnitude can differ by as much as 0.06, corresponding to approximately 60% of the climatological JJA DAOD based on MRC for the Caribbean, and can differ by as much as 0.05, approximately 20% of the climatological JJA DAOD based on MRC for the tropical North Atlantic. This is because total AOD is the only aerosol property constrained by satellite observations through AOD data assimilation in all three aerosol reanalysis products, while speciated AOD is not constrained. This also supports the potential usefulness of MRC, as multi-model-consensus are found to generally be better performers than individual models in aerosol simulations (Sessions et al., 2015; Xian et al., 2019). Despite differences in DAOD magnitude, DAODs of the three reanalysis products correlate significantly over the tropical Atlantic and Caribbean.

2. Each of the three individual reanalyses and the MRC have significant and negative correlations between JJA Caribbean DAOD and seasonal Atlantic Accumulated Cyclone Energy (ACE) (Table 2). High JJA DAOD in the Caribbean is associated with a less conducive environment for hurricane activity as represented by cooler SST, enhanced vertical wind shear, lower mid-level moisture, and by lower Maximum Potential Intensity (MPI) and Genesis Potential Index (GPI) values (Table 3). Pronounced differences in Atlantic TC activity are seen when examining the three seasons with the highest levels of JJA Caribbean DAOD compared with the three seasons with the lowest JJA Caribbean DAOD (Table 4 and Figure 7). About three times as many major hurricanes occurred during the three lowest DAOD seasons (2005, 2011, 2017) compared with the three highest DAOD seasons (2018, 2015, 2014). Atlantic TC activity is also negatively correlated with tropical North Atlantic DAOD but not as significantly as with Caribbean DAOD for possible reasons discussed in the conclusion that follows.

3. High Caribbean DAOD is typically associated with El Niño conditions, however ENSO does not appear to significantly impact the Caribbean DAOD-ACE relationship. The robust DAOD-ACE correlation still holds after removing ENSO's influence via partial correlation analysis.

4. JJA north Atlantic DAOD and the Atlantic Meridional Mode (AMM) are intertwined in the dust-TC relationship. Both the Caribbean and tropical North Atlantic DAODs have strong negative correlations with JJA values of the AMM index (with a stronger correlation for the Caribbean DAOD). Meanwhile, the JJA AMM index correlates significantly with Atlantic ACE. For AMM and DAOD, removing the other in their relationships with ACE considerably reduces the significance of the correlations based on partial correlation analysis. This result supports Evan et al.(2011)'s work, which showed that African dust excited AMM variability on interannual to decadal time scales through radiative forcing of the underlying SST. Consequently, it can be argued that the negative correlation between Caribbean and tropical North Atlantic DAOD and Atlantic TC activity may be a result of forcing of the AMM by African dust.



These results agree with previous studies that showed negative correlations between boreal summer Atlantic dustiness and TC
activity (e.g., Dunion and Velden, 2004; Evan et al., 2006a; Lau and Kim 2007). The correlations obtained in this study,
especially those with Caribbean DAOD, are slightly higher than previous studies, including correlations between boreal
summer dustiness and ACE (Evan et al., 2006a; Lau and Kim 2007) and between JJA dustiness and ENSO (DeFlorio et al.,
2016). We note that the study areas, time periods and study methods are not identical between our study and these previous
studies, implying the usefulness of aerosol reanalyses in climate studies. Various sensitivity tests show that our results are not
sensitive to the definitions of areas for the Caribbean and tropical North Atlantic, the number of composite years used, or the
definition of the dust season (June-July vs. June-August).

Our results also document statistically significant relationships between Atlantic dustiness and large-scale fields (e.g., SST,
vertical wind shear, mid-level moisture and relatively vorticity) and integrated TC indices (e.g., MPI and GPI), which can be
seen as confounding factors in the dust-TC relationship. Exploring the causality of the documented negative correlation
between DAOD and Atlantic TC activity through modeling experiments is beyond the scope of this paper. However in some
modeling studies, radiative forcing of the scattering dust alone could result in an inverse relationship between African dust and
Atlantic TC activity (Dunstone et. al, 2013; Strong et al., 2018; Sobel et al., 2019), along with consistent large-scale fields
associated with TC activity. So it is possible that radiative forcing of the scattering dust is a dominant factor in the inverse
dust-TC relationship on seasonal-interannual and basin-wide scales. After all, aerosol is coupled radiatively with meteorology
in the ERA-Interim dataset, which provides the large-scale atmospheric data used in this study.

The correlations with Atlantic ACE are higher for Caribbean DAOD than for tropical North Atlantic DAOD - a result that was
not documented previously. The differences in the relationships of Atlantic ACE with regional DAOD are potentially due to
several factors:

a)        The large-scale environmental fields investigated herein are better self-correlated between the JJA and ASO seasons
over the Caribbean than over the tropical North Atlantic (Table 4). Therefore, the higher correlation with JJA Caribbean
DAOD, which reflects the large-scale circulation, especially for the low- and mid-level wind flow that modulates how far
African dust can be transported, tends to extend into the ASO peak TC season. As was shown in Saunders et al. (2017), the
strength of the low-level winds in the Caribbean tends to be the most robust diagnostic for seasonal ACE during the peak
months of the Atlantic hurricane season.

b)        ENSO has a large impact on the Caribbean large-scale environment (Gray, 1984). ENSO-forced SST anomalies
typically exhibit very strong persistence from JJA to ASO (Harrison & Larkin, 1998).  As Caribbean JJA DAOD is correlated
with ENSO to some extent, it is also correlated with Atlantic ACE.

c)        There might be a regime shift in the integrated outcome of dust-TC interactions from east to west across the tropical
North Atlantic at climate time scales, with the eastern tropical North Atlantic (e.g., close to the African continent) having a





positive correlation (e.g., since dust emission is often associated with AEWs emerging from Africa; Jones et al., 2003; Karyampudi and Carlson, 1988) while the correlation becomes more negative heading west across the basin.

d)    The relatively larger JJA DAOD variability in the Caribbean (larger relative dynamical range of Caribbean JJA DAOD compared to the tropical north Atlantic JJA DAOD over 2003-2018) could contribute to a higher correlation (Figures 3 and 4).

e)    AOD data quality is comparatively better over the Caribbean than over the tropical north Atlantic (Table 1; Figure 3), given that AOD reanalyses generally perform better over long-range transport regions than they do closer to the aerosol source regions (Xian et al., 2016).

The conclusions drawn from this study are based on 16-year records (2003-2018) of DAOD data from the MRC. We acknowledge that this analysis represents a relatively short timespan for a climate correlation study. Longer-period aerosol

reanalyses with good fidelity are needed for further climate studies. These reanalyses face several challenges including dealing with changes to the network of AOD-observing satellites over time, as well as reasonable simulations and partitioning of aerosol-speciated AODs associated with rare severe aerosol events (e.g., volcanic eruptions) within AOD data assimilation systems.

It is also worth noting that the 2003-2018 DAOD climatology for the Atlantic MDR could be low relative to the extremely

dusty period of the 1980s that has been documented by long-term in-situ dust concentration measurements made in Barbados (Prospero, 2014) and a 24-year eastern North Atlantic dust cover record derived from the Advanced Very High Resolution Radiometer (AVHRR, Evan et al., 2006). Dust concentration records at Barbados (1965-2011) and dust cover determined from AVHRR (1982-2005) both indicate that dust levels over the North Atlantic peaked during the mid-1980s, when tropical Atlantic TC activity was relatively low (e.g., Wang et al., 2012). Our findings are consistent with the negative correlations

reflected in both the earlier dust and the earlier tropical Atlantic TC activity records.

Since we note that JJA/JJ Caribbean DAOD are strongly correlated with large-scale atmospheric fields which are frequently used in seasonal hurricane forecasts, DAOD may be a useful confirmation tool for observed thermodynamic conditions. Most groups issuing seasonal hurricane forecasts provide an update in early August (immediately prior to the peak of the Atlantic hurricane season). Early season (June-July) DAOD, especially over the Caribbean, could be a potential indicator for the

strength of seasonal Atlantic TC activity. While there is very strong agreement between large-scale zonal wind fields (e.g., the June-July-averaged 850 hPa zonal wind over the tropical Atlantic correlates at 0.92 between ERA-Interim and MERRA-2, in which impact of aerosols on radiation and thereby on meteorology is included), there is less agreement with the mid-level moisture field. The correlation between ERA-Interim and MERRA-2 700 hPa RH over the tropical Atlantic averaged over June-July is only 0.72. The DAOD could potentially be used to help clarify the favorability/unfavorability of the




thermodynamic environment, as indicated also by the strong correlation between JJA Caribbean DAOD and JJA and ASO
MPI.

We believe this study, which shows a robust correlation relationship for the integrated dust-atmosphere-ocean system, could
provide a framework to better understand the linkages between DAOD and Atlantic TC activity and how DAOD affects the
large-scale environment of the MDR. This study focuses on seasonal to interannual time scales and provides aspects from a
large-scale point of view. Finer temporal scales (from hours to days) are needed in future studies for cases where African dust
is entrained into TC vortices. These studies would be useful for improved understanding of the potential impacts of dust on
both TC intensity and track.

**Acknowledgments:**
NAAPS-RA development and application and authors P. Xian and J.S. Reid are supported by the Office of Naval Research
Code 322. P. Klotzbach would like to acknowledge a grant from the G. Unger Vetlesen Foundation. P. Colarco is supported
by the NASA Modeling, Analysis, and Prediction program (program manager: David Considine).

**Data Availability:** All data supporting the conclusions of this manuscript are available through the links provided below:
AERONET Version 3 Level 2 data: http://aeronet.gsfc.nasa.gov
AMM index: https://www.esrl.noaa.gov/psd/data/timeseries/monthly/AMM/ammsst.data
CAMSRA AOD: https://www.ecmwf.int/en/research/climate-reanalysis/cams-reanalysis
ENSO index: http://origin.cpc.ncep.noaa.gov/products/analysis_monitoring/ensostuff/ONI_v5.php
ERA-interim monthly means: https://rda.ucar.edu/datasets/ds627.1/
HURDAT2: https://www.aoml.noaa.gov/hrd/hurdat/hurdat2.html
MERRA-2 AOD: https://disc.gsfc.nasa.gov/datasets/M2TMNXAER_V5.12.4/summary?keywords=%22MERRA-2%22
NAAPS RA AOD: https://usgodae.org//cgi-bin/datalist.pl?dset=nrl_naaps_reanalysis&summary=Go
NOAA OI SST V2 data: https://www.esrl.noaa.gov/psd/

**Author contribution:** P. J. K, J. P. D. and P.X. conceived the idea. P.X. and P.J.K performed most of the analysis. M.A.J.
calculated and performed the analyses on MPI and GPI. All authors contributed to the writing and revision of the manuscript.

**Competing interests:** The authors declare that they have no conflict of interest.

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

2017GL072617.




**Tables:**


**Table 1**. Root mean square error of total AOD (left number in each cell) and coarse-mode AOD (right number in each cell) at 550nm from individual aerosol reanalyses, including CAMSRA, MERRA-2, and NAAPS-RA, and the Multi-Reanalysis-Consensus (MRC) verified with AERONET V3L2 monthly data for the 2003-2018 time period. The rank of MRC among all reanalyses in terms of RMSE is also shown. "~" means there are ties in the ranking. The first five sites are located in North Africa or off of the northwestern coast of Africa. The last

four sites are located in or near the Caribbean Sea. AOD time series of MRC and AERONET at Dakar, Capo Verde, Ragged Point and La Parguera sites are presented in Figure 3.

| Site | CAMSRA | MERRA-2 | NAAPS-RA | MRC | Rank of MRC |
|------|--------|---------|----------|-----|-------------|
| Banizoumbou | 0.13 \| 0.17 | 0.10 \| 0.10 | 0.11 \| 0.13 | 0.10 \| 0.11 | ~1 \| 2 |
| Capo Verde | 0.07 \| 0.07 | 0.06 \| 0.05 | 0.06 \| 0.07 | 0.06 \| 0.06 | ~1 \| 2 |
| Dakar | 0.07 \| 0.11 | 0.07 \| 0.07 | 0.08 \| 0.10 | 0.06 \| 0.08 | 1 \| 2 |
| La Laguna | 0.06 \| 0.05 | 0.06 \| 0.05 | 0.06 \| 0.05 | 0.05 \| 0.04 | 1 \| 1 |
| Santa Cruz Tenerife | 0.04 \| 0.04 | 0.04 \| 0.04 | 0.04 \| 0.04 | 0.03 \| 0.03 | 1 \| 1 |
| Ragged Point | 0.05 \| 0.03 | 0.03 \| 0.03 | 0.03 \| 0.03 | 0.04 \| 0.03 | 3 \| ~1 |
| La Parguera | 0.05 \| 0.02 | 0.03 \| 0.02 | 0.03 \| 0.02 | 0.03 \| 0.02 | ~1 \| ~1 |
| Guadeloupe | 0.05 \| 0.05 | 0.05 \| 0.04 | 0.04 \| 0.05 | 0.04 \| 0.04 | ~1 \| ~1 |
| Key Biscayne | 0.05 \| 0.03 | 0.03 \| 0.02 | 0.02 \| 0.02 | 0.03 \| 0.02 | 2 \| ~1 |





**Table 2.** Annual average Atlantic TC activity in the three seasons with the highest JJA Caribbean DAOD (2014, 2015 and 2018) and the three seasons with the lowest JJA Caribbean DAOD (2005, 2011 and 2017). Ratios between the three low and the three high DAOD seasons are also provided. Corresponding numbers for the Caribbean are provided in parentheses next to the total basin-wide numbers.


|  | Tropical Depressions and Named Storms | Named Storms | Hurricanes | Major Hurricanes | Accumulated Cyclone Energy |
|---|---|---|---|---|---|
| Three highest JJA Caribbean DAOD | 12.3 (2.3) | 11.3 (2.3) | 6.0 (0.3) | 2.0 (0.0) | 86 (3) |
| Three lowest JJA Caribbean DAOD | 23.0 (6.7) | 21.3 (6.7) | 10.7 (3.0) | 5.7 (2.7) | 199 (39) |
| Ratio | 1.9 (2.9) | 1.9 (3.0) | 1.8 (9.0) | 2.8 (N/A) | 2.3 (13.0) |





**Table 3.** Correlation matrix between regionally-averaged multi-reanalysis-consensus (MRC) JJA tropical North Atlantic/Caribbean DAOD and 850 hPa U, 200 hPa U, 200 hPa minus 850 hPa U (zonal wind shear), 700 hPa RH, SST, Maximum Potential Intensity (MPI) and Genesis Potential Index (GPI) during JJA and ASO, respectively. Correlations that are statistically significant at the 90% level are highlighted in bold and those with asterisks are statistically significant at the 95% level. 850 hPa relative vorticity is not shown as none of these correlations were statistically significant.

| Environmental Field | JJA tropical North Atlantic / JJA Caribbean | ASO tropical North Atlantic / ASO Caribbean |
|---|---|---|
| 850 hPa U | **-0.63\* / -0.79\*** | 0.03 / **-0.66\*** |
| 200 hPa U | 0.34 **/ 0.81\*** | 0.30 / **0.85\*** |
| 200 minus 850 hPa U | 0.43 **/ 0.83\*** | 0.22 / **0.82\*** |
| 700 hPa RH | -0.43 **/ -0.80\*** | -0.34 / **-0.55\*** |
| SST | **-0.71\* / -0.75\*** | **-0.44** / **-0.79\*** |
| MPI | **-0.74\* / -0.68\*** | **-0.47** / **-0.67\*** |
| GPI | **-0.75\* / -0.76\*** | **-0.62\* / -0.69\*** |

**Table 4.** Correlation matrix between JJA and ASO values of 850 hPa U, 200 hPa U, 200 minus 850 hPa U, 700 hPa RH and SST over the tropical North Atlantic and the Caribbean, respectively. Correlations that are statistically significant at the 90% level are highlighted in bold, and those with asterisks are statistically significant at the 95% level.

| Environmental Field | Tropical North Atlantic | Caribbean |
|---|---|---|
| 850 hPa U | **0.51\*** | **0.84\*** |
| 200 hPa U | **0.71\*** | **0.88\*** |
| 200 minus 850 hPa U | **0.72\*** | **0.90\*** |
| 700 hPa RH | **0.48** | **0.74\*** |
| SST | **0.77\*** | **0.85\*** |





**Table 5:** Correlations of MRC JJA Caribbean DAOD or JJA tropical North Atlantic (TATL) DAOD with JJA or ASO Oceanic Nino Index
(ONI) and the AMM. Correlations that are statistically significant at the 90% level are highlighted in bold. Correlations that are statistically
significant at the 95% level are marked with *.

|  | JJA Caribbean DAOD | | | | JJA TATL DAOD | | | |
|---|---|---|---|---|---|---|---|---|
|  | CAMSRA | MERRA-2 | NAAPS-RA | MRC | CAMSRA | MERRA-2 | NAAPS-RA | MRC |
| JJA ONI | 0.33 | 0.29 | **0.50** | **0.44** | 0.24 | 0.18 | 0.28 | 0.26 |
| ASO ONI | 0.42 | 0.41 | **0.61*** | **0.54*** | 0.38 | 0.30 | 0.42 | 0.41 |
| JJA AMM | **-0.72*** | **-0.70*** | **-0.63*** | **-0.76*** | **-0.58*** | **-0.65*** | -0.36 | **-0.60*** |
| ASO AMM | -0.42 | **-0.45** | -0.30 | **-0.45** | -0.33 | **-0.52*** | 0.01 | -0.33 |

**Table 6.** Partial correlation matrix between MRC JJA Caribbean and tropical North Atlantic (TATL) DAOD and annual Atlantic basin-wide
ACE while controlling for ENSO (using the JJA ONI) and the AMM (using JJA AMM index). Linear correlations (without controlling for
the climate modes) between DAOD and ACE, AMM and ENSO indices and ACE are also listed for comparison purposes. Correlations that
are statistically significant at the 90% level are highlighted in bold, and those with asterisks are statistically significant at the 95% level. Note
that the thresholds for statistical significance are different for partial correlation and linear correlation as the degrees of freedom are different
between the two.

|  | ACE (ctrl. for ENSO) | ACE (ctrl. for AMM) | ACE | ACE (ctrl. for Caribbean DAOD) | ACE (ctrl. for TATL DAOD) |
|---|---|---|---|---|---|
| MRC Caribbean DAOD | **-0.62*** | -0.29 | **-0.61*** | - | - |
| MRC TATL DAOD | -0.39 | -0.07 | -0.41 | - | - |
| AMM index (JJA/ASO) | - | - | **0.59***/**0.45** | 0.26/0.26 | **0.48**/0.37 |
| ONI (JJA/ASO) | - | - | -0.12/-0.20 | 0.20/0.18 | -0.02/-0.05 |


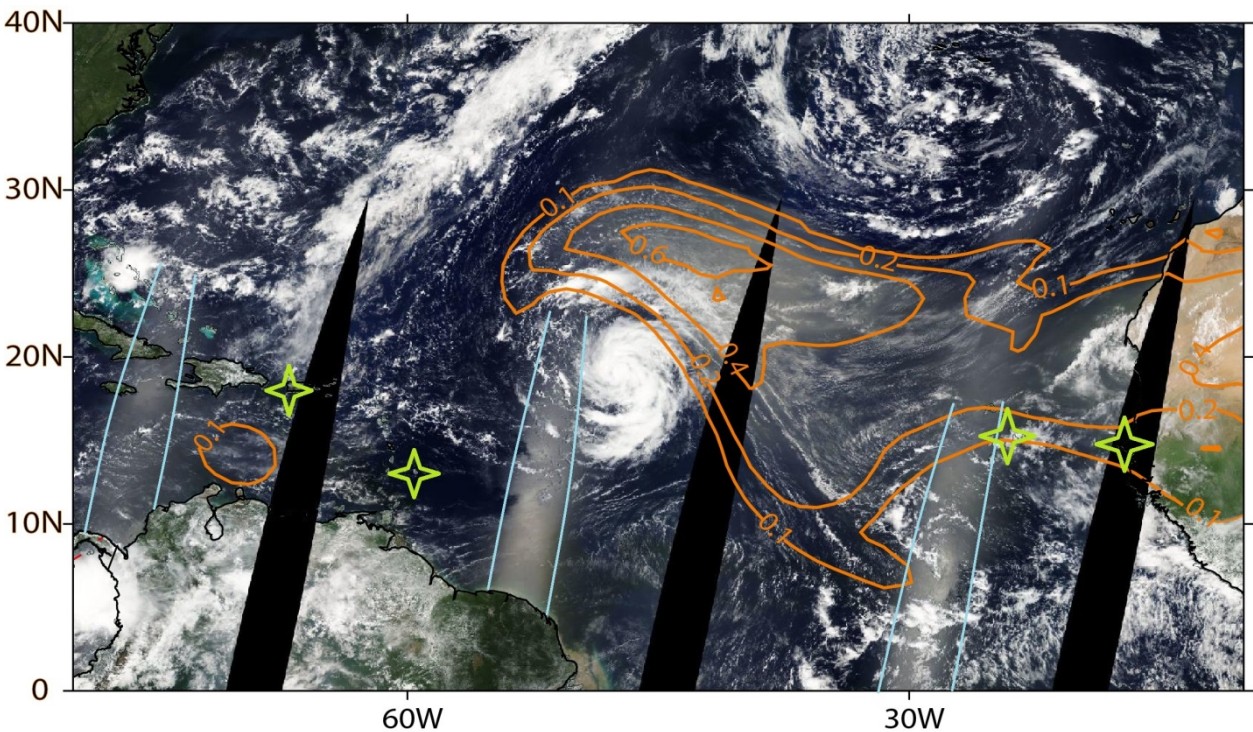

**Figure 1.** True-color Terra MODIS satellite imagery composited on 12 September 2012 and overlaid with NAAPS-RA 550 nm dust aerosol optical depth (DAOD, an approximate measure of total atmospheric column of dust aerosol mass, unitless) isopleths, showing Hurricane Nadine's interaction with the SAL. Nadine is located in the middle of the image. African dust appears as a light transparent brown haze in between the African coast and Nadine and wrapping around the northern periphery of the storm. Note that areas of sunglint (narrow regions between the light blue curves) are similar in color to the dust aerosols, but have the same orientation as the satellite orbits and are located approximately mid-way between satellite coverage gaps (black regions oriented south-southwest to north-northeast). Satellite imagery courtesy of the Moderate Resolution Imaging Spectroradiometer (MODIS) flying on NASA's Terra satellite and available from https://worldview.earthdata.nasa.gov/. The stars in light green represent four sites used for validation purposes, including Dakar, Capo Verde, Ragged Point and La Parguera in order from east to west.


**Figure 2.** Climatological (2003-2018 average) monthly mean DAOD (left column) and the ratio of DAOD to total AOD (right column) for June-October based on the MRC. The middle column shows the climatological 700 hPa RH (color shading) and horizontal wind vectors from ERA-Interim. Black boxes denote the MDR, including the Caribbean (left, 10°-20°N, 85°-60°W) and the tropical North Atlantic (right, 10°-20°N, 60°-20°W, denoted "TATL").


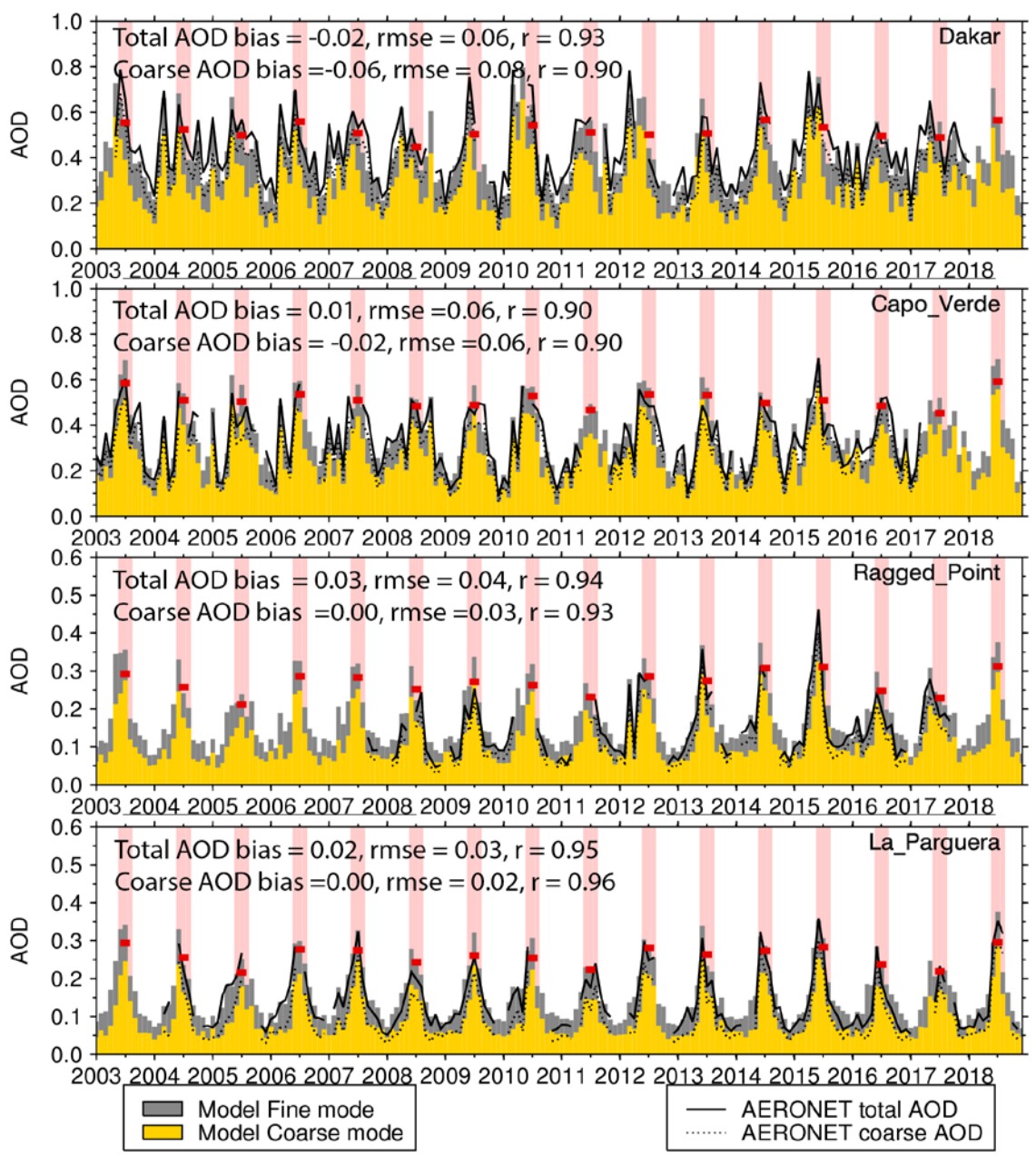

**Figure 3.** Monthly mean Version 3 L2 AERONET and MRC 550nm modal AODs at four African dust-impacted sites: Dakar, Cape Verde, Ragged Point and La Parguera from east to west. JJA are highlighted with pink shading and JJA seasonal average total AOD from MRC are shown by the red bars. Annotations for each time series show bias, RMSE and correlation (*r*) of monthly averages calculated from the MRC.



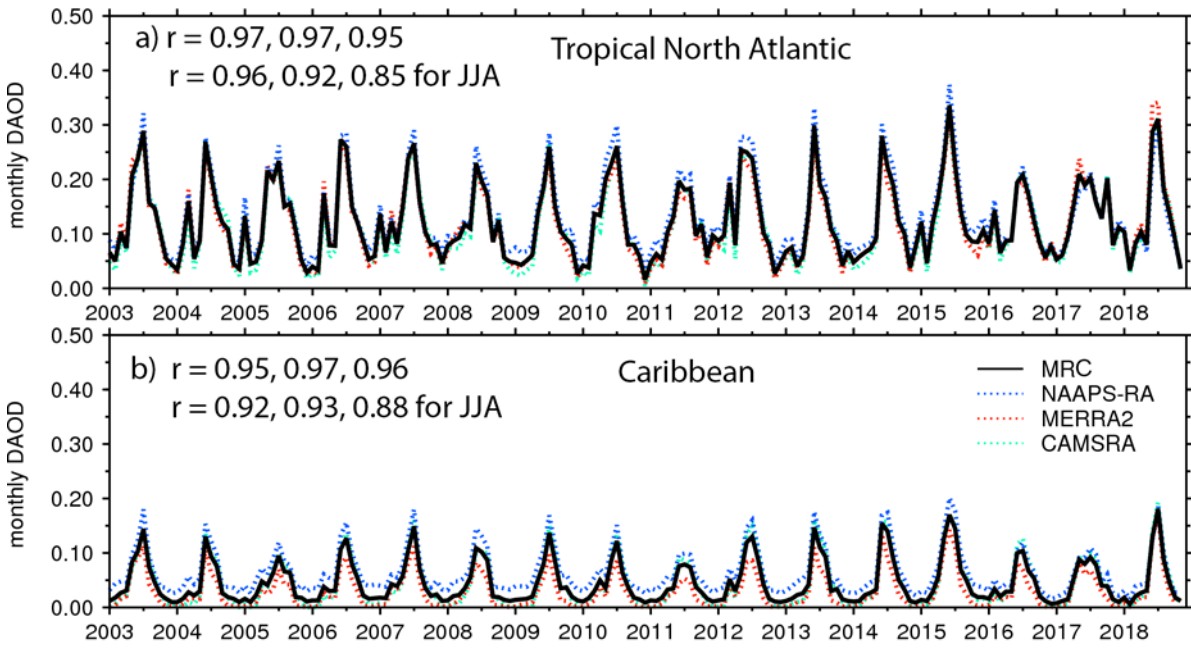


**Figure 4.** Monthly DAOD at 550nm from CAMSRA, MERRA-2, NAAPS-RA and MRC from 2003-2018 for the (a) tropical North Atlantic and the (b) Caribbean. Correlations between CAMSRA and MERRA-2, CAMSRA and NAAPS-RA, MERRA-2 and NAAPS-RA are displayed in sequence for all months and for JJA-only respectively.





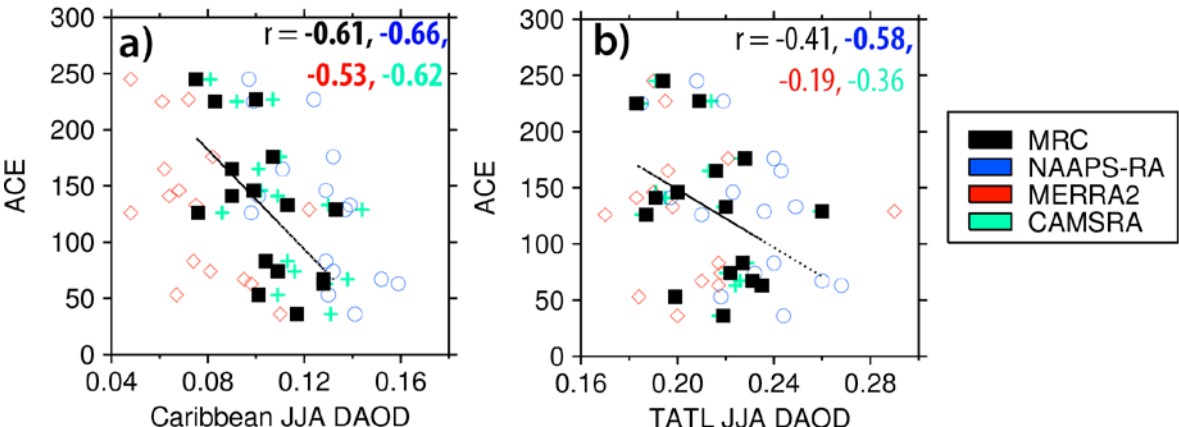


**Figure 5.** Scatterplot showing the relationship between annual Atlantic Accumulated Cyclone Energy (ACE) and JJA region-averaged DAOD from MRC, NAAPS-RA, MERRA-2 and CAMSRA over the (a) Caribbean and (b) the tropical North Atlantic. One unit of ACE equals $10^4$ kt$^2$. Correlations (*r*) between ACE and DAOD are color-coded for different DAOD products and overlaid on each plot, and statistically significant correlations are in bold. The possible causes of the DAOD difference between the reanalysis products are discussed in Section 3.2.




**Figure 6.** Left: MRC JJA composite of DAOD for the three Atlantic hurricane seasons from 2003-2018 with (a) the highest JJA Caribbean
DAOD (2014, 2015 and 2018), (b) the lowest JJA Caribbean DAOD (2005, 2011, and 2017), and (c) the difference between the two (highest
minus lowest). Right: The corresponding JJA composite of 850 hPa wind (vectors) and 700 hPa RH (color shading).


**Figure 7.** Formation locations of Atlantic named storms during the three seasons with (a) the highest Caribbean JJA DAOD (2014, 2015 and 2018) and (b) the lowest Caribbean JJA DAOD (2005, 2011, and 2017). Also displayed are the maximum intensity that each TC reached.

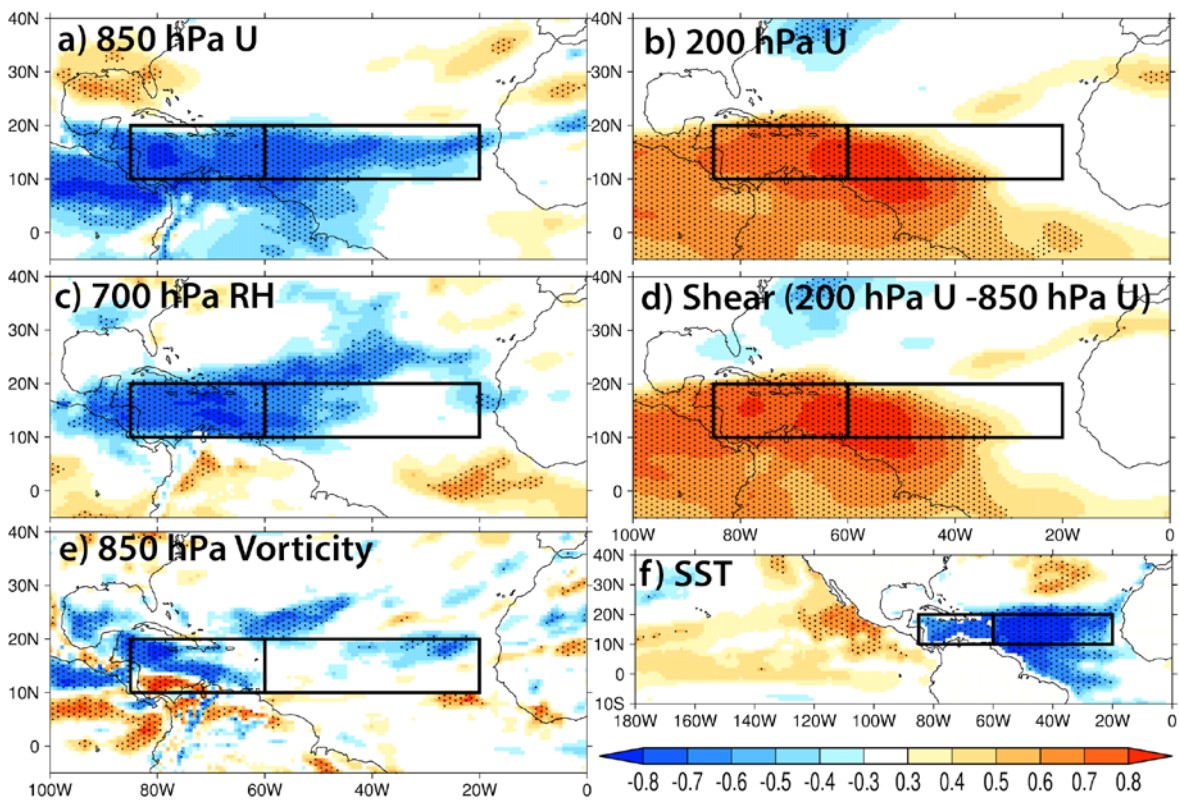

**Figure 8.** Correlation between MRC JJA regionally-averaged DAOD in the Caribbean and JJA (a) 850 hPa U, (b) 200 hPa U, (c) 700 hPa RH, (d) zonal wind shear, (e) 850 hPa relative vorticity and (f) SST. Correlations over the black dotted areas are statistically significant.





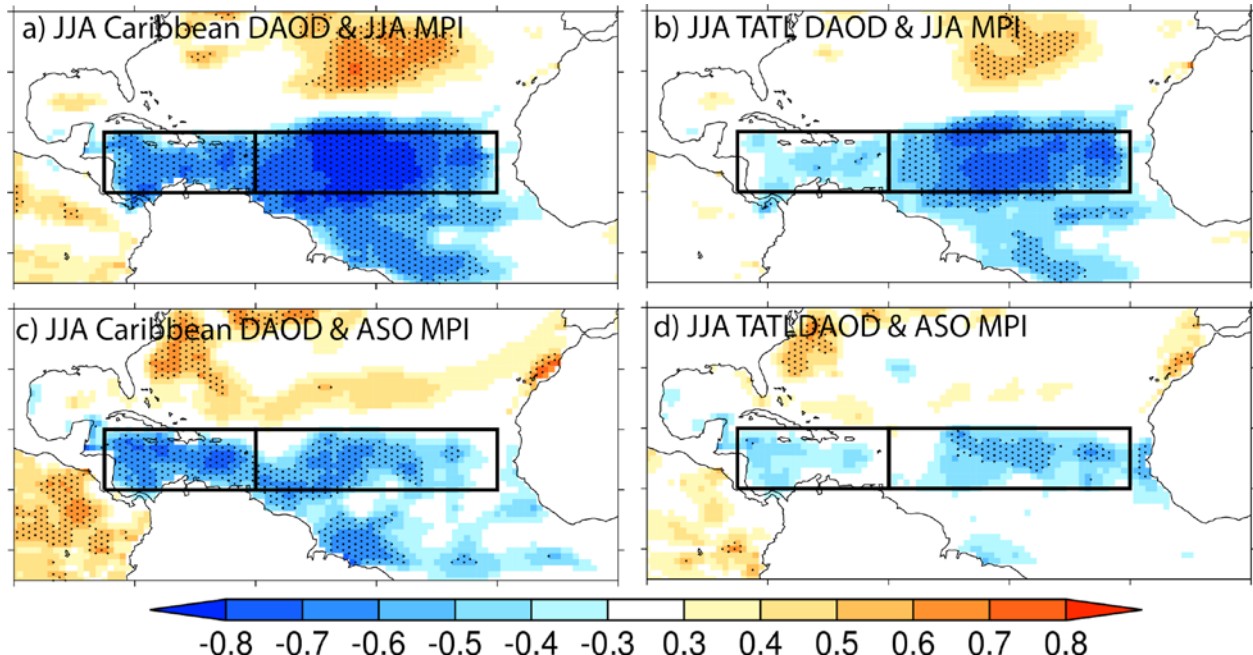

**Figure 9.** Correlation between (a) MRC JJA regionally-averaged DAOD in the Caribbean and JJA Maximum Potential Intensity (MPI), (b) MRC JJA regionally-averaged DAOD in the tropical North Atlantic (TATL) and JJA MPI, (c) JJA Caribbean DAOD and ASO MPI and (d) JJA TATL DAOD and ASO MPI. Correlations over the black dotted areas are statistically significant.





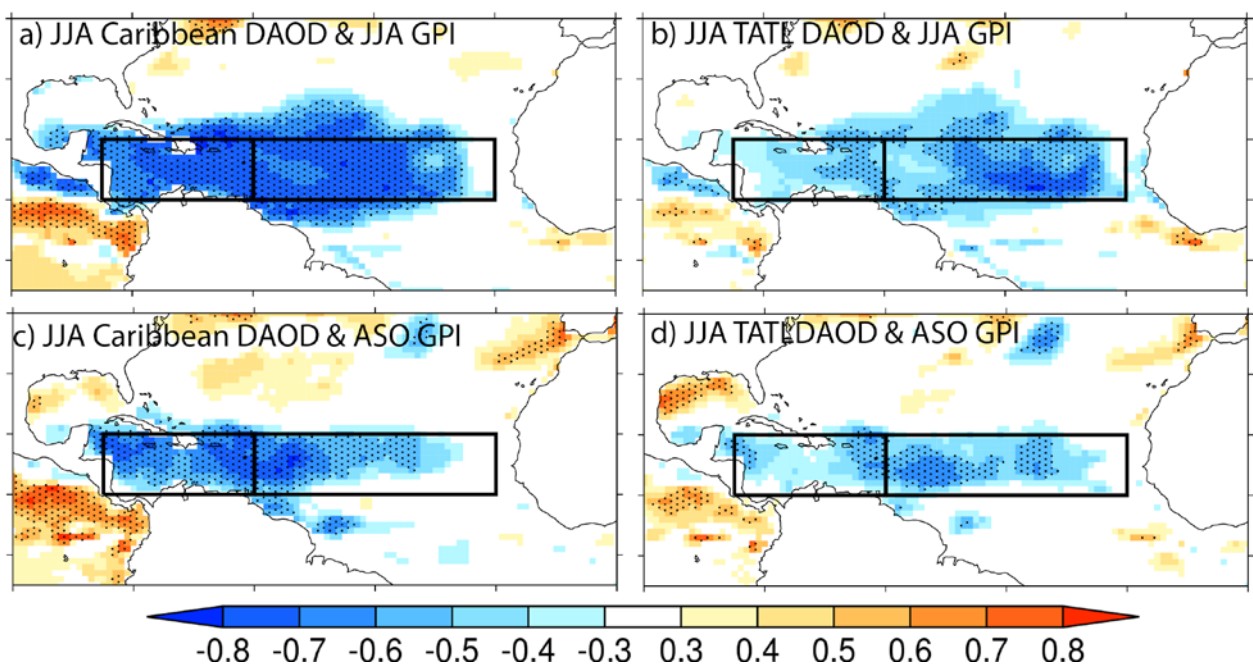

Figure 10. As in Fig. 9, but for the Genesis Potential Index (GPI).