# Peer review of "Activity using Aerosol Optical Depth Reanalyses: 2003-2018 Peng Xian1a, Philip J. Klotzbach2a, Jason P. Dunion3, Matthew A. Janiga1, Jeffrey S. Reid1, Peter R. Colarco4 and Zak Kipling5 1Naval Research Laboratory"

_Atmospheric Chemistry and Physics, 2020_

## Referee Comment (RC1) · Anonymous Referee #2 · 20 Aug 2020

Revisiting the Relationship between Atlantic Dust and Tropical Cyclone Activity using Aerosol Optical Depth Reanalyses: 2003-2018

Peng Xian, Philip J. Klotzbach, Jason P. Dunion, Matthew A. Janiga, Jeffrey S. Reid , Peter R. Colarco and Zak Kipling

Presented in this study is a comprehensive investigation of the relationship between Saharan dust and the activities of tropical cyclones over the Atlantic. Multiple datasets have been used for the analysis and the data analysis, as far as I can see, seems to have been done diligently. The paper is generally well written and has an extensive list of previous published papers. Overall the paper makes a useful contribution to the

research of dust impact on climate.

The paper presents a lot of information, such that the reader cannot easily focus on a particular issue and draw a specific conclusion. In this aspect, I believe the paper can be written better. In particular, a strategy for the data analysis should be presented, so that the reader can have an overview of what hypotheses are being tested and how they will be tested. Such a strategy should be based on, not only the data available but also the hypothesis how Saharan dust might influence TC activities. For example, while Section 2 states data and methods, there is actually hardly any description of the methods. Related to this problem is the lack of a cohesive and overarching interpretation of the results.

Otherwise, I think the paper is fine.
* * *

---

## Referee Comment (RC2) · Anonymous Referee #1 · 8 Sep 2020

The authors have demonstrated that the use of a multi-model ensemble methodology can be applied to the Aerosol reanalyses to produce a more useful and higher quality product that spans a long enough period for statistical analysis (MRC). The quality and utility of this approach is not overly surprising given the performance of the ICAP multi-model ensemble already described in the literature for NWP, but it demonstrated clearly in the comparisons against AERONET observations. As the impact of mineral dust on the development of Atlantic Hurricanes has been a long standing question, with a variety of conflicting analyses and hypotheses, the application of the this long-running and high quality MRC data is a valuable contribution to the question at hand. Because the MRC has both these qualities, the results are particularly convincing. The

main result, in figure 5, that dust AOD is negatively correlated with Atlantic Hurricanes, and the subsequent analysis that this is probably dominated not by the dust itself, but by the circulation patterns associated with the dusty years, is a useful and important result.

---

## Author Response (AR1)

Reply to review comments

RC1 "The authors have demonstrated that the use of a multi-model ensemble methodology can be applied to the Aerosol reanalyses to produce a more useful and higher quality product that spans a long enough period for statistical analysis (MRC). The quality and utility of this approach is not overly surprising given the performance of the ICAP multi-model ensemble already described in the literature for NWP, but it demonstrated clearly in the comparisons against AERONET observations. As the impact of mineral dust on the development of Atlantic Hurricanes has been a long standing question, with a variety of conflicting analyses and hypotheses, the application of the this longrunning and high quality MRC data is a valuable contribution to the question at hand. Because the MRC has both these qualities, the results are particularly convincing. The main result, in figure 5, that dust AOD is negatively correlated with Atlantic Hurricanes, and the subsequent analysis that this is probably dominated not by the dust itself, but by the circulation patterns associated with the dusty years, is a useful and important result."

AC1

*We would like to thank the anonymous reviewer for his/her comments on this paper and appreciation of this work. Based on this comment, there is no need for revision. However we have added a method subsection to describe the strategy for data analysis as suggested by RC2.*

RC2

"Presented in this study is a comprehensive investigation of the relationship between Saharan dust and the activities of tropical cyclones over the Atlantic. Multiple datasets have been used for the analysis and the data analysis, as far as I can see, seems to have been done diligently. The paper is generally well written and has an extensive list of previous published papers. Overall the paper makes a useful contribution to the research of dust impact on climate.

The paper presents a lot of information, such that the reader cannot easily focus on a particular issue and draw a specific conclusion. In this aspect, I believe the paper can be written better. In particular, a strategy for the data analysis should be presented, so that the reader can have an overview of what hypotheses are being tested and how they will be tested. Such a strategy should be based on, not only the data available but also the hypothesis how Saharan dust might influence TC activities. For example, while Section 2 states data and methods, there is actually hardly any description of the methods. Related to this problem is the lack of a cohesive and overarching interpretation of the results.

Otherwise, I think the paper is fine."

*AC2*

*Thank you very much for the comment, which helps to improve the readability and flow of the manuscript. In response, and to describe the strategy for the data analysis, we have added a "Methods" subsection under section 2. Besides adding the data analysis strategy, we have moved the original subsection "2.7 Statistical correlation calculations and significance tests" to the end of this "method" subsection, as it fits better here. We have also moved the few sentences describing the study regions from the introduction section to the new "Methods" subsection for the same reason.*

*With the added description of the strategy for data analysis and method, we feel that a cohesive and overarching interpretation of the results is also achieved. Thank you again for the helpful comment. The new subsection reads as follows*

"2.1 Methods

Regardless of the underlying mechanisms, as there are contradicting mechanisms proposed in different studies, the goal of this study is to examine if there is a robust and statistically significant relationship between African dust and Atlantic TC activity on seasonal to interannual time scales. We also examine if there are confounding factors, for example, meteorological conditions and climate modes that co-vary with dust and hence influence TC activity.

We use dust AOD (DAOD) to represent Atlantic dust levels. Three aerosol reanalysis products, and their consensus DAOD are used in order to increase the fidelity of the analysis result, given that multi-model-consensus typically has been shown to have better data quality in prior assessments (Sessions et al., 2015; Xian et al., 2019). Various TC count indices and Accumulated Cyclone Energy (ACE) (Bell et al. 2000), defined in the next section, are utilized to represent TC activity.

The Atlantic Main Development Region (MDR) (e.g., Goldenberg et al., 2001), including the Caribbean (10-20°N, 85-60°W) and the tropical North Atlantic (10-20°N, 60-20°W), are the focus regions for this study (see also Figure 2 for a spatial representation of the two subregions). Most previous studies of dust impacts on TC activity have focused on the tropical North Atlantic or regions closer to the African continent (e.g., Karyampudi and Pierce, 2002; Bretl et al. 2015; Pan et al., 2018) where DAOD is relatively high. However significant dust pulses can also be transported into the Caribbean. We therefore expand our study area to explore the potential impacts of high levels of dust in the Caribbean on Atlantic TC activity. This allows us to explore regional differences in the dust-TC relationship. Statistical relationships between DAOD and TC activity over the MDR are investigated using the three aerosol reanalyses and multi-reanalysis-consensus (MRC). The results obtained herein also help us assess the potential of using DAOD to aid in future Atlantic seasonal hurricane forecasts.

The correlations between variables of interest are based on the Pearson correlation coefficient. Statistical significance is assessed at the 95% level using a two-tailed Student's t-test. Correlations >= 0.51 are statistically significant given that a 16-year time period (e.g., 2003-2018) is investigated here. For partial correlation analysis, partial correlations >=0.55 are statistically significant at the 95% level with 13 degrees of freedom. The criteria for statistical significance with various degrees of freedom can also be obtained at: https://www.esrl.noaa.gov/psd/data/correlation/significance.html."

Marked-up manuscript includes

1. *A newly added "Methods" subsection under section 2. Besides adding the data analysis strategy, we have moved the original subsection "2.7 Statistical correlation calculations and significance tests" to the end of this "method" subsection, as it fits better here. We have also moved the few sentences describing the study regions from the introduction section to the new "Methods" subsection for the same reason.*
2. *New sequencing numbers for other subsections in section 2.*
3. *We have also made some very minor changes throughout the text, e.g. typos, and language.*

[revised manuscript text omitted]

---

## Author Response (AR2)

**Dear Editor,**

Thank you very much for your editing work on our paper entitled "Revisiting the Relationship between Atlantic Dust and Tropical Cyclone Activity using Aerosol Optical Depth Reanalyses: 2003–2018". Thank you also for your suggestion of emphasizing the key findings of the paper in the abstract and conclusions, which would make the key findings standing out more. In response to your suggestion, we have added one concluding sentence in the abstract and a similar concluding sentence in the last paragraph of the conclusions section. We hope you find this technical correction is satisfactory. The concluding sentence in the abstract reads

"Overall, DAOD in both the tropical Atlantic and Caribbean is negatively correlated with Atlantic hurricane frequency and intensity, with stronger correlations in the Caribbean than farther east in the tropical North Atlantic."

In addition, we have revised the reference format in the paper following ACP's reference guidance in response to the advice from your editorial team. The manuscript with marked up changes are attached below.

Sincerely,

Peng

Peng Xian, PhD Atmospheric Properties and Effects Naval Research Laboratory Marine Meteorology Division 7 Grace Hopper Avenue, Stop 2 Monterey, California 93940 831-656-4803 FAX 831-656-4769 peng.xian@nrlmry.navy.mil

**Revisiting the Relationship between Atlantic Dust and Tropical Cyclone Activity using Aerosol Optical Depth Reanalyses: 2003-2018**

- Peng Xian1a, Philip J. Klotzbach2a, Jason P. Dunion3, Matthew A. Janiga1, Jeffrey S. Reid1, Peter
   R. Colarco4 and Zak Kipling5
- 7 1Naval Research Laboratory, Monterey, CA, USA.

[revised manuscript text omitted]
 Sanarah dust on numcane genesis. Journal                                                                                                                                                                                                                                                                                                                                                                                                                                                                                                                                                                                                                                                                                                                                                                                                                                                                                                                                                                                                                                                                                                                                                                                                                                                                                                                                                                                                                                                                                                                                                                                                                                                                                                                                                                                                                                                                                                                                                                                                                                                                                                                                                                                                                                                                                                                                                                                                                                                                                                                                                                                                                                                                                                                                           | Formatted: Font: Not Italic |
| 852 | bit $\frac{1}{200}$ or $\frac{1}{200}$ $\frac{1}{200}$ $\frac{1}{400}$ $\frac{1}{200}$ $\frac{1}{200}$ $\frac{1}{400}$ $\frac{1}{200}$ $\frac{1}{200}$ $\frac{1}{400}$ $\frac{1}{4000}$ $\frac{1}{400}$ $\frac{1}{400$ |                             |
| 055 | Puchard V. Bandlas C. A. da Silva A. Darmanov, A. S. Calarao, B. B. Covinderaju, B.C. at                                                                                                                                                                                                                                                                                                                                                                                                                                                                                                                                                                                                                                                                                                                                                                                                                                                                                                                                                                                                                                                                                                                                                                                                                                                                                                                                                                                                                                                                                                                                                                                                                                                                                                                                                                                                                                                                                                                                                                                                                                                                                                                                                                                                                                                                                                                                                                                                                                                                                                                                                                                                                                                                                                                                               |                             |
| 855 | al (2017) The MERRA-2 Aerosol Reanalysis 1980 onward Part II: Evaluation and case                                                                                                                                                                                                                                                                                                                                                                                                                                                                                                                                                                                                                                                                                                                                                                                                                                                                                                                                                                                                                                                                                                                                                                                                                                                                                                                                                                                                                                                                                                                                                                                                                                                                                                                                                                                                                                                                                                                                                                                                                                                                                                                                                                                                                                                                                                                                                                                                                                                                                                                                                                                                                                                                                                                                                      |                             |
| 856 | studies Journal of Climate 30(17) 6851-6872 https://doi.org/10.1175/ICL LD-16-0613.1.2017                                                                                                                                                                                                                                                                                                                                                                                                                                                                                                                                                                                                                                                                                                                                                                                                                                                                                                                                                                                                                                                                                                                                                                                                                                                                                                                                                                                                                                                                                                                                                                                                                                                                                                                                                                                                                                                                                                                                                                                                                                                                                                                                                                                                                                                                                                                                                                                                                                                                                                                                                                                                                                                                                                                                              | Formattade Cant: Nat Italia |
| 857 | Carlson, T. N., & And Prospero, J. M. (1972): The large-scale movement of Saharan air                                                                                                                                                                                                                                                                                                                                                                                                                                                                                                                                                                                                                                                                                                                                                                                                                                                                                                                                                                                                                                                                                                                                                                                                                                                                                                                                                                                                                                                                                                                                                                                                                                                                                                                                                                                                                                                                                                                                                                                                                                                                                                                                                                                                                                                                                                                                                                                                                                                                                                                                                                                                                                                                                                                                                  |                             |
| 858 | outbreaks over the northern equatorial Atlantic. Journal of Applied Meteorology. 11(2), 283–                                                                                                                                                                                                                                                                                                                                                                                                                                                                                                                                                                                                                                                                                                                                                                                                                                                                                                                                                                                                                                                                                                                                                                                                                                                                                                                                                                                                                                                                                                                                                                                                                                                                                                                                                                                                                                                                                                                                                                                                                                                                                                                                                                                                                                                                                                                                                                                                                                                                                                                                                                                                                                                                                                                                           | Formatted: No underline     |
| 859 | 297. https://doi.org/10.1175/1520-0450(1972)011<0283:TLSMOS>2.0.CO;2, 1972,                                                                                                                                                                                                                                                                                                                                                                                                                                                                                                                                                                                                                                                                                                                                                                                                                                                                                                                                                                                                                                                                                                                                                                                                                                                                                                                                                                                                                                                                                                                                                                                                                                                                                                                                                                                                                                                                                                                                                                                                                                                                                                                                                                                                                                                                                                                                                                                                                                                                                                                                                                                                                                                                                                                                                            | Formatted: Font: Not Italic |
| 860 | Camargo, S. J., Emanuel, K. A. & and Sobel, A. H. (2007).: Use of a genesis potential index to                                                                                                                                                                                                                                                                                                                                                                                                                                                                                                                                                                                                                                                                                                                                                                                                                                                                                                                                                                                                                                                                                                                                                                                                                                                                                                                                                                                                                                                                                                                                                                                                                                                                                                                                                                                                                                                                                                                                                                                                                                                                                                                                                                                                                                                                                                                                                                                                                                                                                                                                                                                                                                                                                                                                         | Formatted: No underline     |
| 861 | diagnose ENSO effects on tropical cyclone genesis. J. Clim. 20, 4819–4834, 2007.                                                                                                                                                                                                                                                                                                                                                                                                                                                                                                                                                                                                                                                                                                                                                                                                                                                                                                                                                                                                                                                                                                                                                                                                                                                                                                                                                                                                                                                                                                                                                                                                                                                                                                                                                                                                                                                                                                                                                                                                                                                                                                                                                                                                                                                                                                                                                                                                                                                                                                                                                                                                                                                                                                                                                       | Formatted: Font: Not Italic |
| 862 | Chin, M., Rood, R., Lin, S., Muller J., & and Thompson, A. (2000): Atmospheric sulfur cycle                                                                                                                                                                                                                                                                                                                                                                                                                                                                                                                                                                                                                                                                                                                                                                                                                                                                                                                                                                                                                                                                                                                                                                                                                                                                                                                                                                                                                                                                                                                                                                                                                                                                                                                                                                                                                                                                                                                                                                                                                                                                                                                                                                                                                                                                                                                                                                                                                                                                                                                                                                                                                                                                                                                                 | Formatted: Font: Not Bold   |
| 863 | simulated in the global model GOCART: Model description and global properties. Journal of                                                                                                                                                                                                                                                                                                                                                                                                                                                                                                                                                                                                                                                                                                                                                                                                                                                                                                                                                                                                                                                                                                                                                                                                                                                                                                                                                                                                                                                                                                                                                                                                                                                                                                                                                                                                                                                                                                                                                                                                                                                                                                                                                                                                                                                                                                                                                                                                                                                                                                                                                                                                                                                                                                                                              | Formatted: Font: Not Italic |
| 864 | Geophysical Research: Atmospheres, $105(D20)$ , $24,671-24,687$ .                                                                                                                                                                                                                                                                                                                                                                                                                                                                                                                                                                                                                                                                                                                                                                                                                                                                                                                                                                                                                                                                                                                                                                                                                                                                                                                                                                                                                                                                                                                                                                                                                                                                                                                                                                                                                                                                                                                                                                                                                                                                                                                                                                                                                                                                                                                                                                                                                                                                                                                                                                                                                                                                                                                                                                      |                             |
| 865 | https://doi.org/10.1029/2000JD900384, 2000.                                                                                                                                                                                                                                                                                                                                                                                                                                                                                                                                                                                                                                                                                                                                                                                                                                                                                                                                                                                                                                                                                                                                                                                                                                                                                                                                                                                                                                                                                                                                                                                                                                                                                                                                                                                                                                                                                                                                                                                                                                                                                                                                                                                                                                                                                                                                                                                                                                                                                                                                                                                                                                                                                                                                                                                     |                             |
| 866 | (2002) Scheren duct transport to the Caribberg during PDIDE 2. Transport vertical restillar                                                                                                                                                                                                                                                                                                                                                                                                                                                                                                                                                                                                                                                                                                                                                                                                                                                                                                                                                                                                                                                                                                                                                                                                                                                                                                                                                                                                                                                                                                                                                                                                                                                                                                                                                                                                                                                                                                                                                                                                                                                                                                                                                                                                                                                                                                                                                                                                                                                                                                                                                                                                                                                                                                                                            |                             |
| 867 | (2003). Sanaran dust transport to the Carlobean during PKIDE: 2. Transport, vertical profiles,                                                                                                                                                                                                                                                                                                                                                                                                                                                                                                                                                                                                                                                                                                                                                                                                                                                                                                                                                                                                                                                                                                                                                                                                                                                                                                                                                                                                                                                                                                                                                                                                                                                                                                                                                                                                                                                                                                                                                                                                                                                                                                                                                                                                                                                                                                                                                                                                                                                                                                                                                                                                                                                                                                                                         |                             |
| 868 | and deposition in simulations of in situ and remote sensing observations. Journal of Geophysical                                                                                                                                                                                                                                                                                                                                                                                                                                                                                                                                                                                                                                                                                                                                                                                                                                                                                                                                                                                                                                                                                                                                                                                                                                                                                                                                                                                                                                                                                                                                                                                                                                                                                                                                                                                                                                                                                                                                                                                                                                                                                                                                                                                                                                                                                                                                                                                                                                                                                                                                                                                                                                                                                                                                       | Formatted: Font: Not Italic |
| 869 | Colarao B. da Silva A. Chin M. Stand Diabl. T. (2010) : Online simulations of global acrossel                                                                                                                                                                                                                                                                                                                                                                                                                                                                                                                                                                                                                                                                                                                                                                                                                                                                                                                                                                                                                                                                                                                                                                                                                                                                                                                                                                                                                                                                                                                                                                                                                                                                                                                                                                                                                                                                                                                                                                                                                                                                                                                                                                                                                                                                                                                                                                                                                                                                                                                                                                                                                                                                                                                                          | Formatted: No underline     |
| 871 | distributions in the NASA GEOS-4 model and comparisons to satellite and ground-based aerosol                                                                                                                                                                                                                                                                                                                                                                                                                                                                                                                                                                                                                                                                                                                                                                                                                                                                                                                                                                                                                                                                                                                                                                                                                                                                                                                                                                                                                                                                                                                                                                                                                                                                                                                                                                                                                                                                                                                                                                                                                                                                                                                                                                                                                                                                                                                                                                                                                                                                                                                                                                                                                                                                                                                                           |                             |
| 872 | optical depth. Journal of Geophysical Research: Atmospheres, 115(D14207).                                                                                                                                                                                                                                                                                                                                                                                                                                                                                                                                                                                                                                                                                                                                                                                                                                                                                                                                                                                                                                                                                                                                                                                                                                                                                                                                                                                                                                                                                                                                                                                                                                                                                                                                                                                                                                                                                                                                                                                                                                                                                                                                                                                                                                                                                                                                                                                                                                                                                                                                                                                                                                                                                                                                                              | Formatted: Font: Not Italic |
| 873 | https://doi.org/10.1029/2009JD012820, 2010.                                                                                                                                                                                                                                                                                                                                                                                                                                                                                                                                                                                                                                                                                                                                                                                                                                                                                                                                                                                                                                                                                                                                                                                                                                                                                                                                                                                                                                                                                                                                                                                                                                                                                                                                                                                                                                                                                                                                                                                                                                                                                                                                                                                                                                                                                                                                                                                                                                                                                                                                                                                                                                                                                                                                                                                            | Formatted, No underline     |
| 874 | Dee, D. P., Uppala, S. M., Simmons, A. J., Berrisford, P., Poli, P., Kobayashi, S., Andrae, U., et                                                                                                                                                                                                                                                                                                                                                                                                                                                                                                                                                                                                                                                                                                                                                                                                                                                                                                                                                                                                                                                                                                                                                                                                                                                                                                                                                                                                                                                                                                                                                                                                                                                                                                                                                                                                                                                                                                                                                                                                                                                                                                                                                                                                                                                                                                                                                                                                                                                                                                                                                                                                                                                                                                                                     | romatted: No undenine       |
| 875 | al. (2011) The ERA-Interim reanalysis: configuration and performance of the data assimilation                                                                                                                                                                                                                                                                                                                                                                                                                                                                                                                                                                                                                                                                                                                                                                                                                                                                                                                                                                                                                                                                                                                                                                                                                                                                                                                                                                                                                                                                                                                                                                                                                                                                                                                                                                                                                                                                                                                                                                                                                                                                                                                                                                                                                                                                                                                                                                                                                                                                                                                                                                                                                                                                                                                                          |                             |
| 876 | system. Quarterly Journal of the Royal Meteorological Society, 137(656), 553-597.                                                                                                                                                                                                                                                                                                                                                                                                                                                                                                                                                                                                                                                                                                                                                                                                                                                                                                                                                                                                                                                                                                                                                                                                                                                                                                                                                                                                                                                                                                                                                                                                                                                                                                                                                                                                                                                                                                                                                                                                                                                                                                                                                                                                                                                                                                                                                                                                                                                                                                                                                                                                                                                                                                                                                      | Formatted: Font: Not Italic |
| 877 | https://doi.org/10.1002/qj.828, 2011.                                                                                                                                                                                                                                                                                                                                                                                                                                                                                                                                                                                                                                                                                                                                                                                                                                                                                                                                                                                                                                                                                                                                                                                                                                                                                                                                                                                                                                                                                                                                                                                                                                                                                                                                                                                                                                                                                                                                                                                                                                                                                                                                                                                                                                                                                                                                                                                                                                                                                                                                                                                                                                                                                                                                                                                                  | Formatted: No underline     |

| 878 | DeFlorio, M. J., Goodwin, I. D., Cayan, D. R., Miller, A. J., Ghan, S. J., Pierce, D. W., Russell,       |                             |
|-----|----------------------------------------------------------------------------------------------------------|-----------------------------|
| 879 | L. M., et al. (2016) Interannual modulation of subtropical Atlantic boreal summer dust                   |                             |
| 880 | variability by ENSO. Climate Dynamics, 46(1), 585-599. https://doi.org/10.1007/s00382-015-               | Formatted: Font: Not Italic |
| 881 | 2600-7, 2016.                                                                                            | Formatted: No underline     |
| 882 | DeMott, P., Sassen, K., Poellot, M., Baumgardner, D., Rogers, D., Brooks, S. D., Prenni, A. J.,          |                             |
| 883 | & and Kreidenweis, S. M. (2003): African dust aerosols as atmospheric ice nuclei. Geophysical | Formatted: Font: Not Italic |
| 884 | Research Letters, 30, 1732. https://doi.org/10.1029/2003GL017410, 2003.                           | Formatted: No underline     |
| 885 | Doherty, O. M., Riemer, N., & and Hameed, S. (2008): Saharan mineral dust transport into the  |                             |
| 886 | Caribbean: Observed atmospheric controls and trends, Journal of Geophysical Research:                    | Formatted: Font: Not Italic |
| 887 | Atmospheres, 113(D07211). https://doi.org/10.1029/2007JD009171, 2008.                                    | Formatted: No underline     |
| 888 | Dunion, J. P., (2011) Re-writing the climatology of the tropical North Atlantic and Caribbean            |                             |
| 889 | Sea atmosphere. Journal of Climate, 24(3), 893–908. https://doi.org/10.1175/2010JCLI3496.1,              | Formatted: Font: Not Italic |
| 890 | 2011.                                                                                                    | Formatted: No underline     |
| 891 | Dunion, J. P., & and and Marron, C. S. (2008).: A reexamination of the Jordan mean tropical              |                             |
| 892 | sounding based on awareness of the Saharan air layer: Results from 2002. Journal of Climate,             | Formatted: Font: Not Italic |
| 893 | 21(20), 5242–5253. https://doi.org/10.1175/2008JCLI1868.1, 2008.                                  | Formatted: No underline     |
| 894 | Dunion, J. P., & and Velden, C. S. (2004) .: The impact of the Saharan Air Layer on Atlantic             |                             |
| 895 | tropical cyclone activity. Bulletin of the American Meteorological Society, 85(3), 353-365.              | Formatted: Font: Not Italic |
| 896 | https://doi.org/10.1175/BAMS-85-3-353, 2004.                                                             | Formatted: No underline     |
| 897 | Emanuel, K. A., & and Nolan, D. S. ( : 2004). Tropical cyclone activity and global climate. In:   |                             |
| 898 | Proc. of 26th Conference on Hurricanes and Tropical Meteorology, pp. 240–241, American                   | Formatted: Font: Not Italic |
| 899 | Meteorological